# Bayesian Methods for Constraint Inference in Reinforcement Learning

**Dimitris Papadimitriou**  *dimitri@berkeley.edu*
*UC Berkeley*

**Usman Anwar**  *ua237@cam.ac.uk*
*University of Cambridge*

**Daniel S. Brown**  *daniel.s.brown@utah.edu*
*University of Utah*

**Reviewed on OpenReview:** *https://openreview.net/forum?id=oRjk5V9eDp&referrer=*

## Abstract

Learning constraints from demonstrations provides a natural and efficient way to improve the safety of AI systems; however, prior work only considers learning a single, point-estimate of the constraints. By contrast, we consider the problem of inferring constraints from demonstrations using a Bayesian perspective. We propose *Bayesian Inverse Constraint Reinforcement Learning* (BICRL), a novel approach that infers a posterior probability distribution over constraints from demonstrated trajectories. The main advantages of BICRL, compared to prior constraint inference algorithms, are *(1)* the freedom to infer constraints from partial trajectories and even from disjoint state-action pairs, *(2)* the ability to infer constraints from suboptimal demonstrations and in stochastic environments, and *(3)* the opportunity to use the posterior distribution over constraints in order to implement active learning and robust policy optimization techniques. We show that BICRL outperforms pre-existing constraint learning approaches, leading to more accurate constraint inference and consequently safer policies. We further propose Hierarchical BICRL that infers constraints locally in sub-spaces of the entire domain and then composes global constraint estimates leading to accurate and computationally efficient constraint estimation.

## 1 Introduction

Reinforcement Learning (RL) algorithms have proven effective in providing policies for a wide range of dynamic decision making tasks (Mnih et al., 2013; Polydoros & Nalpantidis, 2017; Charpentier et al., 2021). However, manually specifying a reward function in an environment to encourage an agent to perform a specific task is a nontrivial process. To alleviate this issue, *Inverse Reinforcement Learning* (IRL) aims at inferring a reward function by observing the behavior of an expert agent performing a specified task (Russell, 1998). While many different IRL approaches have been proposed to infer non-degenerate reward functions from a finite set of expert trajectories (Arora & Doshi, 2021), in many cases it may not be necessary to infer the entire reward function, since partial information regarding the reward function might be available. For example, in safety critical applications such as robotic surgery (Lanfranco et al., 2004) and autonomous driving (Shafaei et al., 2018), the basic goal of a task may be known (e.g. move the robot end effector to a particular position using minimal energy or minimize travel time while avoiding collisions), but there may be user-specific constraints that are unknown (e.g. constraints on the amount of pressure applied during surgery or the proximity to people or other cars when driving). In these cases, we desire algorithms that can infer the unknown constraints by observing demonstrations from an expert in the environment.

Prior work has considered constraint learning from demonstrations in a *maximum likelihood* setting, without considering or utilizing a representation of uncertainty of the constraints. Chou et al. (2018) reason that

trajectories that result in lower cumulative costs than the demonstrated ones must be associated with constraints. Based on the same notion, Scobee & Sastry (2019) propose a greedy method to add constraints in a MDP so that the expert demonstrated trajectories are more likely under that choice of constraints. Finally, Anwar et al. (2020) extend the aforementioned method to continuous state spaces with unknown transition models. Park et al. (2020) use a Bayesian non-parametric approach to estimate a sequence of subgoals and corresponding constraints from demonstrations; however, they only obtain the MAP solution and assume the demonstrator never violates constraints. By contrast, we infer a full Bayesian posterior distribution over constraints while considering demonstrators that are imperfect and may sometimes accidentally violate constraints. Maintaining a belief distribution over the location and likelihood of constraints is important for many downstream tasks such as active query synthesis (Settles, 2009), Bayesian robust optimization (Brown et al., 2020a; Javed et al., 2021) and safe exploration (García & Fernández, 2015).

In this work, we formulate the constraint inference problem using a Bayesian perspective and demonstrate that our approach has a number of advantages over prior work. For example, as opposed to the maximum likelihood counterpart (Scobee & Sastry, 2019) which requires full expert trajectory demonstrations, our approach works with partial trajectories (even disjoint state-action pairs) as well as full trajectories. Our proposed Bayesian approach also allows us to estimate uncertainty over the true constraints, enabling active constraint learning where the agent can query the expert for specific state actions or partial demonstrations. One benefit of learning constraints is that they are naturally disentangled from the task reward. Building on this idea, we demonstrate the ability to learn constraints in a hierarchical manner: learning constraints from local demonstrations to accelerate and improve global constraint inference via a divide and conquer approach. Finally, by extending and generalizing prior work by Ramachandran & Amir (2007a), we prove that our Bayesian constraint inference approach has desirable rapid mixing guarantees.

## 2 Related Work

**Constraint Inference:** Constraint inference has generally been studied (Chou et al., 2018; 2020) with focus on inferring unknown constraints of specific types e.g. geometric (D'Arpino & Shah, 2017; Subramani et al., 2018), sequential (Pardowitz et al., 2005) or convex (Miryoosefi et al., 2019). As an important innovation, Scobee & Sastry (2019) formulated the problem of inferring general constraints in the framework of inverse reinforcement learning and provided a greedy algorithm to infer constraints in deterministic tabular environments. Their proposed approach has since been extended to work with stochastic demonstrations (McPherson et al., 2021) and continuous state spaces (Stocking et al., 2022). Anwar et al. (2020) developed an alternative approach, with focus on scaling to high dimensional continuous state space environments. Chou et al. (2022) in parallel have developed an approach to learn chance-constraints using Gaussian processes for motion planning.

**Preference Learning and Inverse RL:** Constraint inference can also be viewed as a special case of preference learning (Christiano et al., 2017). Preference learning focuses on learning the preference order over outcomes through means of ratings (Daniel et al., 2014), comparisons (Christiano et al., 2017; Sadigh et al., 2017), human interventions (MacGlashan et al., 2017; Hadfield-Menell et al., 2017a) or other forms of human feedback (Jeon et al., 2020). Demonstrations (Ziebart et al., 2008; Finn et al., 2016; Brown et al., 2020c) are a particularly popular form of feedback. In imitation learning (Ross et al., 2011), provided demonstrations are considered optimal and the AI agent is trained to reproduce the demonstrated behavior. However, imitation learning is known to suffer from issues such as lack of robustness to noisy observations (Reddy et al., 2020), distribution shift (Ross et al., 2011) and fragile learning (Zolna et al., 2019). Inverse reinforcement learning (Russell, 1998; Ng et al., 2000; Abbeel & Ng, 2004) avoids some of the issues of imitation learning by learning an explicit reward function from demonstrations first and using regularizers (Finn et al., 2016; Fu et al., 2018), priors (Ramachandran & Amir, 2007b; Michini & How, 2012; Jeon et al., 2018) or robust optimization (Javed et al., 2021) to generalize faithfully. Bayesian IRL (Ramachandran & Amir, 2007b; Brown et al., 2020a;b; Chan & van der Schaar, 2020), in particular, attempts to learn a distribution over possible reward functions.

**Reward Function Design:** Our work also connects with the wider literature on safe reward function design which focuses on minimizing or avoiding *side effects* (Turner et al., 2020) due to reward misspecification. The approaches in this area either manually design a reward function regularizer that inhibits the tendency of RL

agent to act destructively in the environment or learn such a regularizer from information leaked or provided by humans. Turner et al. (2020) use attainable utility preservation as a regularizer which is closely linked to the idea of reachability (Krakovna et al., 2018). Shah et al. (2019) learn the regularizer term by ensuring that the world state in the final state of a demonstration trajectory is maintained. Hadfield-Menell et al. (2017b) consider the given reward function as true reward function only on the given environment and then use Bayesian inference to infer the correct reward function when the environment undergoes any change.

**Safe Reinforcement Learning:** In safe reinforcement learning (Garcıa & Fernández, 2015), the objective is to learn a policy that maximizes the given reward while ensuring minimal constraint violations. In most of the prior works (Garcıa & Fernández, 2015; Achiam et al., 2017; Tessler et al., 2018; Calian et al., 2020; Srinivasan et al., 2020), it has been assumed that the constraint set is known and the focus has been on improving algorithms to learn improved constraint abiding policies efficiently. Yang et al. (2021) is a notable exception to this trend which assumes that constraints are specified through natural language.

## 3 Bayesian Constraint Inference

### 3.1 Preliminaries

Constrained Reinforcement Learning (CRL) (Garcıa & Fernández, 2015), is generally studied in the context of Constrained Markov Decision Processes (CMDP). A CMDP, $M_C$, is a tuple $(S, A, P, R, C, \gamma)$, where $S$ denotes the state space and $A$ the action space. We denote the transition probability $P : S \times A \to S$ from a state $s$ following action $a$ with $P(s'|s, a)$. The transition dynamics are considered known throughout this paper. We denote with $R : S \times A \to \mathbb{R}$ the reward function, with $C$ the set of constraints and with $\gamma \in (0, 1)$ the discount factor. If the state space is discrete with $|S| = n$, then the constraint set $C \in \{0, 1\}^n$ can be modeled as an $n$-ary binary product where $C[j] = 1$ means that the state $j \in \{1, \ldots, n\}$ is a constraint state. Further, we use the indicator function $\mathbb{I}_C$ to denote membership over the constraint set. Denoting the action policy with $\pi$, the CRL objective can be written as follows

$$\max_{\pi} \quad \mathbb{E}_{a \sim \pi} \left[ \sum_{t=1}^{T} \gamma^t R(s, a) \right] \quad \text{s.t.} \quad \mathbb{E}_{a \sim \pi} \left[ \sum_{t=1}^{T} \mathbb{I}_C(s, a) \right] = 0. \tag{1}$$

One way to solve Eq. (1) is by formulating the Lagrangian of the optimization problem and solving the resulting min-max problem

$$\min_{r_p \leq 0} \quad \max_{\pi} \quad \mathbb{E}_{a \sim \pi} \left[ \sum_{t=1}^{T} \gamma^t R(s, a) \right] + r_p \left( \mathbb{E}_{a \sim \pi} \left[ \sum_{t=1}^{T} \mathbb{I}_C(s, a) \right] \right), \tag{2}$$

where $r_p \in (-\infty, 0]$ denotes the Lagrange multiplier. Intuitively, the Lagrange multiplier $r_p$ can be interpreted as the penalty an agent will incur in terms of reward for violating a constraint. Prior work by Paternain et al. (2019), shows that the problem has zero duality gap and hence the solutions of the aforementioned two problems are equivalent. We leverage this fact to pose the constraint inference problem as the inverse of Problem (2). This novel perspective helps us utilize off-the-shelf efficient RL solvers to obtain an optimal policy from Problem (2) for fixed values of $r_p$. That allows us to formulate the constraint inference task as a Bayesian optimization problem over the unknown parameters, without solving a min-max problem. In contrast, prior works (Scobee & Sastry, 2019; Anwar et al., 2020) which formulate the constraint inference problem as the inverse of Problem (1), require the use of constrained RL solvers which are generally much less stable and efficient.

### 3.2 Problem Statement

Bayesian constraint inference is the problem of inferring a distribution over possible constraint sets given $M_C \setminus C$, which denotes the *nominal* MDP with unknown constraints, and a set of demonstrations. Denote the set of demonstrations with $\mathcal{D}$, where $\mathcal{D} = \{\xi_1, \xi_2, \ldots\}$ is composed of a number of trajectories, with each individual trajectory being denoted with $\xi_i = \{(s_1^i, a_1^i), (s_2^i, a_2^i), \ldots\}$. Furthermore, as opposed to prior works

(Scobee & Sastry, 2019; Anwar et al., 2020), we leverage the fact that problems (1) and (2) are equivalent and hence we use the formulation in (2) for constraint inference. This novel perspective, has multiple benefits: *(1)* learning constraints does not require making any modifications to the model of the environment (for example there is no need in modifying the action space in particular states as done in Scobee & Sastry (2019)); *(2)* it allows for learning demonstrator's "risk tolerance" level through learning of $r_p$; and *(3)* this in turn allows the use of standard, and more stable, RL algorithms to solve for the CMDP in the place of CRL algorithms, which are often difficult to tune (Anwar et al., 2020).

### 3.3   Bayesian Constraint Inference Algorithm

In our formulation, an agent can take an action that leads to a constraint state in which case the agent incurs a penalty reward of $r_p < 0$, which is incorporated in the reward function. More specifically, when the agent takes action $a$ and transitions to state $s \in C$ the observed reward is $r_p$. For a transition in a state $s \notin C$ the accrued nominal reward is $r = R(s, a)$. We will be referring to the nominal MDP that is augmented with a constraint set $C$ and a penalty reward $r_p$ as $M_{C,r_p}$ which is defined by $(S, A, P, R, C, r_p, \gamma)$. In this formulation, the entries of the nominal reward vector specified by $C$ have been modified to take the value of $r_p$. With this modified reward function, we can use a MDP solver and obtain an optimal policy for any choice of constraint configuration and penalty reward. Inspired by the classic Bayesian IRL (BIRL) framework (Ramachandran & Amir, 2007a), we propose a modification of the Grid Walk algorithm (Vempala, 2005) to jointly perform Markov Chain Monte Carlo (MCMC) sampling over the constraint set $C$ and the penalty reward $r_p$, as detailed in the next section.

In what follows, we detail our Bayesian constraint inference method called Bayesian Inverse Constraint Reinforcement Learning (BICRL). The basic concept behind our Bayesian method follows the approach proposed in Ramachandran & Amir (2007a) and can be summarized in the following steps: *(1)* we sample a candidate solution, in this case a candidate constraint and a penalty reward, from the neighborhood of the current solution; *(2)* we compare the likelihood functions of the demonstrations under this proposal and the current solution and probabilistically accept the candidate solution based on that comparison. As in the Metropolis-Hastings algorithm for Markov Chain Monte Carlo methods, we also allow for randomly accepting proposals even if they are not associated with a higher likelihood to enhance exploration in the Markov Chain. In our implementation, at each iteration we either sample a candidate constraint or a penalty reward, which is an approach reminiscent of alternating optimization algorithms. This choice can is controlled by the user specified sampling frequency $f_r$. For instance, if $f_r = 20$ then we sample constraints 20 times more often than penalty rewards. When sampling constraints, we select a random index $j$ from $\{1, \ldots, n\}$ and only change the constraint status of state $j$: $C'[j] = \neg C[j]$, $C'[i] = C[i], \forall i \neq j$. For the penalty reward we sample $r_p$ from a Gaussian proposal distribution. The main computational burden of the algorithm, lies in the value iteration method that is called in each of the $K$ iterations to evaluate the likelihood given the proposal.

We compute the likelihood of a sample by assuming a Boltzmann-type choice model (Ziebart et al., 2008). Under this model, the likelihood of a trajectory $\xi$ of length $m$ is given by

$$\mathcal{L}(C, r_p) \coloneqq P(\xi | C, r_p) = \prod_{i=1}^{m} \frac{e^{\beta Q^*(s_i, a_i)}}{Z_i}, \tag{3}$$

where $Z_i$ is the partition function and $\beta \in [0, \infty)$ is the inverse of the temperature parameter. Assuming a prior distribution over constraints and penalties $P(C, r_p)$, the posterior distribution is given by

$$P(C, r_p | \xi) = \frac{P(\xi | C, r_p) P(C, r_p)}{P(\xi)}. \tag{4}$$

We choose an uninformative prior in our experiments, but given some domain knowledge informative priors can also be incorporated. The detailed process of sampling from the posterior distribution using BICRL can be seen in Algorithm 1.

The Maximum a Posteriori (MAP) estimates for the constraints and the penalty reward can be obtained as

$$C_{\text{MAP}}, r_{p\text{MAP}} = \arg \max_{C, r_p} P(C, r_p | \xi), \tag{5}$$

---

**Algorithm 1** BICRL

---

1: **Parameters:** Number of iterations $K$, penalty reward sampling frequency $f_r$, standard deviation $\sigma$
2: Randomly sample $C \in \{0, 1\}^n$
3: Randomly sample $r_p \in \mathbb{R}$
4: $chain_C[0] = C$
5: $chain_{r_p}[0] = r_p$
6: Compute $Q^*$ on $M_{C,r_p}$
7: **for** $i = 1, \ldots, K$ **do**
8:     **if** $(i \bmod f_r)!=0$ **then**
9:         Randomly sample state $j$ from $\{1, \ldots, n\}$
10:        Set $C'[j] = \neg C[j]$, $r'_p = r_p$
11:    **else**
12:        Set $r'_p = r_p + \mathcal{N}(0, \sigma)$, $C' = C$
13:    Compute $Q^*$ on $M_{C',r'_p}$
14:    **if** $\log \mathcal{L}(C', r'_p) \geq \log \mathcal{L}(C, r_p)$ **then**
15:        Set $C = C'$, $r_p = r'_p$
16:    **else**
17:        Set $C = C'$, $r_p = r'_p$ w.p. $\mathcal{L}(C', r'_p)/\mathcal{L}(C, r_p)$
18:    $chain_C[i] = C$
19:    $chain_{r_p}[i] = r_p$
20: **Return** $chain_C$, $chain_{r_p}$

---

and the Expected a Posteriori (EAP) estimates as

$$C_{\text{EAP}}, r_{p\text{EAP}} = \mathbb{E}_{C,r_p \sim P(C,r_p|\xi)}[C, r_p|\xi]. \tag{6}$$

The MAP and EAP estimates will be used to evaluate the performance of BICRL. Although the MAP estimates provide the required information to evaluate the classification and possibly obtain new policies, the EAP estimates complement them by quantifying the estimation uncertainty.

### 3.4   Theoretical Properties

One common concern of MCMC algorithms is the speed of convergence. In Appendix D we prove that in the special case of a uniform prior, the Markov chain induced by BICRL mixes rapidly to within $\epsilon$ of the equilibrium distribution in a number of steps equal to $O(N^2 \log 1/\epsilon)$. Our proof follows closely the work of Applegate & Kannan (1991). Our proof is similar to that of Ramachandran & Amir (2007a); however, as we show in the Appendix, the proof in Ramachandran & Amir (2007a) relies on a simplified likelihood function that can lead to pathologically trivial maximum a posteriori reward functions. By contrast, we extend the work of Ramachandran & Amir (2007a) by proving rapid mixing for both Bayesian IRL and BICRL when using the true Boltzmann likelihood function.

### 3.5   Active Constraint Learning

One of the benefits of using a Bayesian approach to infer constraints is the quantification of the uncertainty, or confidence, in the estimation. Safety critical applications require safety guarantees during the deployment of an agent. In that direction, we are interested in mechanisms that can improve the agent's confidence regarding constraint inference. We examine the utility of a simple active learning acquisition function which is based on the variance of the constraint estimates. One of the benefits of BICRL lies in that it does not require entire trajectory demonstrations but specific state action demonstrations suffice. We propose an active learning method in which the agent can query the demonstrator $K_Q$ times for $K_D$ specific state demonstrations associated with high uncertainty in the current estimation. Between queries BICRL is run for $K_A$ iterations. The outline of this process is summarized in Algorithm 2.

At every iteration of the active learning algorithm, the BICRL Algorithm is called to provide a new estimate of the constraints. To expedite the process, initialization in BICRL can be set using a warm start. More specifically, after the first iteration, each of the subsequent calls to BICRL uses the MAP solution of the previous iteration as the constraint and penalty reward initialization (lines 2 and 3 in Algorithm 1).

Despite its simplicity, this active learning approach yields significant improvement and highlights the active learning benefits that can be gained due to the Bayesian nature of the approach. As active learning of constraints is not the primary focus of this work, we leave the investigation of more sophisticated approaches, such as seeking expert samples on states in backward reachable sets of highly uncertain states, to future work.

---

**Algorithm 2** Active Constraint Learning

---

1: **Parameters:** Number of iterations $K_Q$, $K_D$, $K_A$
2: **for** $i = 1, \ldots, K_Q$ **do**
3:     Run BICRL for $K_A$ iterations
4:     Compute $var(C[i])$, $\forall\ i$
5:     Select state $i^* = \mathrm{argmax}_i\ var(C[i])$
6:     Query demonstrator for $K_D$ state $i^*$ demonstrations
7:     Add demonstrations to $\mathcal{D}$
8: **Return** $chain_C$, $chain_{r_p}$

---

## 4 Deterministic Environments

Most recent work in constraint inference in IRL is based on a Maximum Likelihood framework (Scobee & Sastry, 2019; Stocking et al., 2022). Before introducing the main body of our results, we carry out simulations in deterministic state space grid world environments to compare our method to the Greedy Iterative Constraint Inference (GICI) method proposed by Scobee & Sastry (2019). GICI iteratively learns constraints by computing the most likely state, among the ones not appearing in the expert demonstrations, to appear in trajectories obtained by optimizing the current set of learned constraints. This state is then added to the set of learned constraints. GICI, like our method, assumes knowledge of the nominal reward function which refers to the rewards of the constraint free states. One of the restrictions of GICI is that it assumes deterministic transition dynamics and hence in this section we restrict BICRL in deterministic MDPs for a fair comparison.

### 4.1 Classification Performance and Convergence Rate

We first demonstrate the performance of our method in the discrete grid world environment shown in Figure 1a. The goal state, which is marked with a green flag at the top left, is associated with a known reward of 2 while each other constraint-free state is associated with a known reward of $-1$. The environment includes 12 constraint states that can be seen in Figure 1a colored in red. Each constraint state is associated with an unknown penalty reward of $r_p = -10$. We synthesize expert trajectories using a Boltzmann policy

$$\pi(a|s) \propto e^{\beta Q^*(s,a)}, \tag{7}$$

with $\beta = 1$. We provide to our learning algorithm 100 trajectories from the demonstrator, each having as a starting state the bottom left corner, designated with a blue flag. Figure 1a shows the state visitation frequencies of those trajectories. We run BICRL for $K = 2000$ iterations and we sample constraints using a penalty reward sampling frequency $f_r$ of 50. Our Bayesian method correctly identifies the majority of the actual constraints. Figure 1b shows the mean predictions for the constraints with values approaching 1.0 corresponding to high belief of that state being constrained. Mean values close to 0.5 designate states with high uncertainty regarding their constraint status. Given the modeling in BICRL, these values are essentially the parameters of Bernoulli random variables that model the likelihood of constraint existence. The algorithm further manages to infer the penalty reward $r_p$ returning an estimated mean value of $-9.96$. As expected, the agent demonstrates high uncertainty in areas that are far away from the expert demonstrations, like the bottom right section of the grid.

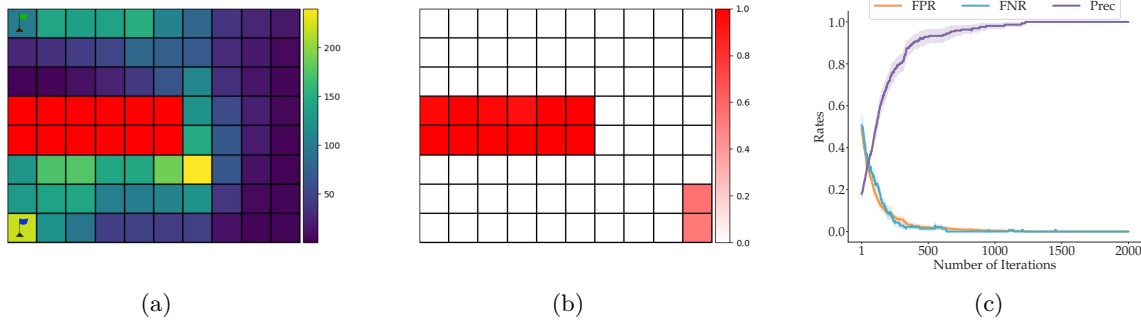

(a)        (b)        (c)

Figure 1: (1a) True constraints (red) along with expert state visitation frequencies. (1b) EAP constraint estimates obtained from BICRL. (1c) Classification rates of MAP constraint estimates. Results averaged over 10 independent experiments.

To quantify the convergence rate and the performance of BICRL, we further provide a plot in Figure 1c of the False Positive Rate (FPR), the False Negative Rate (FNR) and the Precision (Prec) of the MAP estimates at each BICRL iteration. We average the rates over 10 independent runs of BICRL, each using 100 new expert demonstrations. The true constraints are the ones specified in Figure 1a. The rates are not necessarily monotonic, as at each iteration of BICRL, we allow sub-optimal propositions to be accepted to enhance exploration in the Markov Chain. As the number of iterations increases, the MAP estimates tend towards the true constraints.

### 4.2   Bayesian vs Maximum Likelihood Estimation

To further quantify the benefits of our method, we also compare it with GICI. For the same environment as in Figure 1a, we compare the classification performance of both methods in Table 1.

Table 1: False Positive, False Negative and Precision classification rates for GICI and BICRL for varying levels of transition dynamics noise. Results averaged over 10 runs.

| | GICI | | | BICRL | | |
|---|---|---|---|---|---|---|
| $\epsilon$ | FPR | FNR | Precision | FPR | FNR | Precision |
| 0.0 | 0.02 | 0.08 | 0.89 | 0.02 | 0.0 | 0.88 |
| 0.01 | 0.01 | 0.46 | 0.88 | 0.02 | 0.0 | 0.92 |
| 0.05 | 0.0 | 0.83 | 0.90 | 0.0 | 0.0 | 0.99 |

For these simulations, we utilized 100 expert trajectories obtained using a Boltzmann policy with $\beta = 1$. The detailed parameters of the simulation can be found in Appendix A.1. GICI utilizes a KL-divergence based stopping criterion with which it terminates when the distribution of the trajectories under the inferred constraints is within a threshold of the distribution of the expert trajectories. We tuned this criterion accordingly to improve classification results. In the Appendix we include the classification performance of GICI for a grid of KL divergence stopping criterion choices. We average the classification results over 10 simulations. For deterministic transition dynamics, for which $\epsilon$ is 0.0, BICRL and GICI show comparable performance, with BICRL returning fewer false negatives. A low False Negative Rate in the estimation can lead to the acquisition of significantly safer policies. Furthermore, it must be noted that properly tuning the KL criterion requires knowledge of the number of constraints which in reality is not available. In addition, BICRL estimates the entire posterior distribution that can be utilized in active learning tasks. Although this section is focused on deterministic dynamics, we further perform simulations for stochastic environments with low levels of noise, namely for $\epsilon = 0.01$ and $\epsilon = 0.05$. It is evident that even with little noise, BICRL outperforms GICI, which can be attributed to the fact that noise in demonstrations enhances exploration and periodically leads to constraint violation.

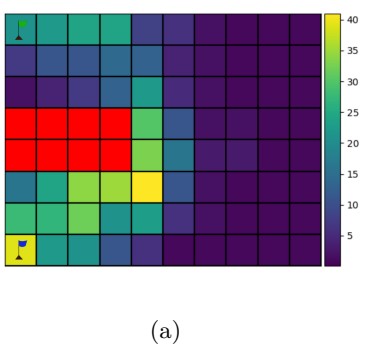
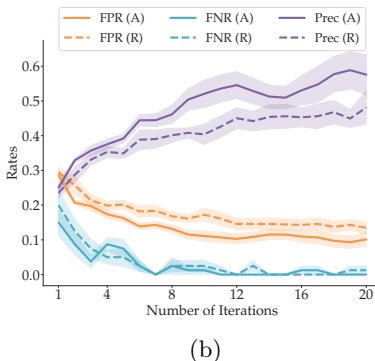

(a)                                                                      (b)

Figure 2: (2a) True constraints (red) along with expert state visitation frequencies. (2b) Classification rates for Active (A) and Random (R) learning methods. Results averaged over 10 independent experiments.

### 4.3 Active Learning

Another advantage of BICRL over GICI, is having access to a posterior distribution of the constraints which inherently allows for active learning. Querying the expert for specific state-action pair demonstrations can allow for faster and more accurate inference. To obtain intuition about the significance of active learning, we compare the active learning method outlined in Algorithm 2 with a random approach that each time randomly selects a state to query the expert for demonstrations. The MDP environment used is shown in Figure 2a. In each experiment, we have a set $\mathcal{D}$ of 20 initial expert demonstrations and we use the active and the random method to query the expert $K_Q = 20$ times for $K_D = 20$ state-action pair demonstrations at a particular state each time with $K_A = 100$. After each iteration $i = 1, \ldots, K_Q$ of the active and random learning methods, we compute the FPR, FNR and Precision.

We deliberately restricted to few expert demonstrations and imposed a smaller constraint set in this MDP in order to increase uncertainty especially on the states on the right side of the grid, as very few trajectories now pass through that region. Figure 2b contains the classification rates averaged over 10 independent experiments. The active learning method outperforms the random one especially in the case of false positives and precision. In these simulations, we assumed there was a budget of 20 iterations for the active learning algorithm. As the number of active learning iterations increases, classification accuracy naturally further improves.

## 5 BICRL Motivation

### 5.1 Comparison With Other Approaches

In this section, we motivate BICRL by showing that the decoupling of constraints and the corresponding penalty rewards is indeed needed for accurate constraint inference. In that direction, we compare BICRL to three alternative methods. The simplest approach in inferring constraint states associated with low rewards is via Bayesian Inverse Reinforcement Learning (BIRL) (Ramachandran & Amir, 2007a). In this case however, we do not assume knowledge of a nominal reward function but infer the entire reward vector instead. In the other two methods, we assume knowledge of the nominal reward function and on top of that we infer the penalty rewards $\mathbf{r}_p \in \mathbb{R}^n$, that are now state dependent, without including in the MCMC sampling the indicator variable set $C$ to designate constraint states. These approaches can essentially be considered as an implementation of BIRL with knowledge of the nominal rewards. In the first variation, called Bayesian Continuous Penalty Reward (BCPR), we assume that $\mathbf{r}_p$ takes on continuous values while in Bayesian Discrete Penalty Reward (BDPR) we assume they are discrete. The detailed algorithms for these two methods, as well the details of the experiments, can be seen in the Appendix.

To compare the above alternatives with BICRL, we use four different MDP environments as seen in Figure 3. Each $\text{MDP}_i, i = 1, 2, 3, 4$ is associated with $k_i$ constraints, $k_1 = 14$, $k_2 = 16$, $k_3 = 10$, $k_4 = 6$, depicted in

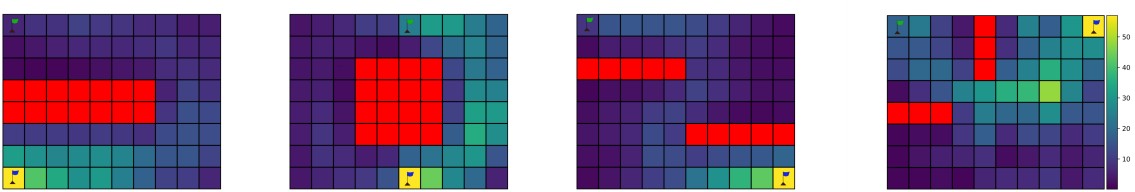

Figure 3: The four grid world environments (from left to right $\mathrm{MDP}_i, i = 1, \ldots, 4$) used to evaluate BICRL. Red squares denote constraint states while the colormap quantifies the state visitation frequencies of the expert trajectories. Start states are denoted with a blue flag and goal states with a green.

red. In each case, an expert provides noisy demonstrations from a start to a goal state which are designated with blue and green flags, respectively. The dynamics are considered to be stochastic in the MDPs. More specifically, when the agent tries to move in a particular direction there is an $\epsilon$ probability that the agent will move to a neighboring state instead. Stochastic dynamics, in addition to providing a more general framework to the deterministic ones, can be seen as a more accurate depiction of real world robot transitions that are not always perfect due to sensor errors, actuator errors or interference from the surrounding environment. For the remainder of this paper we will assume that the transition dynamics are stochastic.

We first compare the False Positive, False Negative and Precision classification rates that we obtain from the MAP estimates of the four methods as shown in Figure 4a. For BIRL, BDPR and BCPR we classify the $k$ states with the lowest $r_p$s (from the vector of $\mathbf{r}_p$s) as constraint states, where $k$ is the number of constraints in the original MDP. It should be noted that assuming knowledge of the actual number of constraint states gives a significant advantage to the three comparison methods. Furthermore, in Figure 4b we report the average number of constraint violations per trajectory for each method. These trajectories are acquired from an optimal policy that is obtained for the MDPs with the inferred constraints and penalty rewards and evaluated on the true MDPs. For each MDP we carry out 10 independent constraint inference estimations using the four algorithms and for each of those we obtain 100 optimal trajectories to evaluate the average constraint violation.

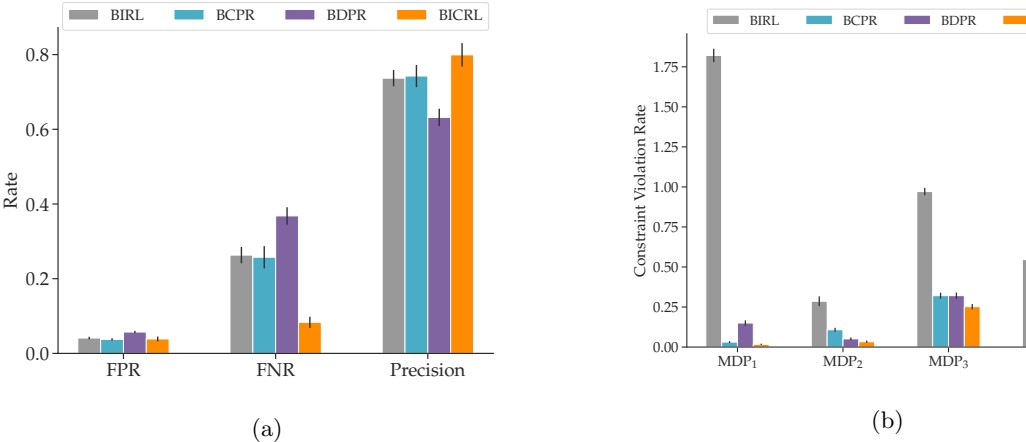

(a)                                                                 (b)

Figure 4: (a) Classification results of BIRL, BCPR, BDPR and BICRL. Results averaged over the four MDPs and 10 independent simulations. (b) Average constraint violation for the four MDPs. Results averaged over 10 independent simulations.

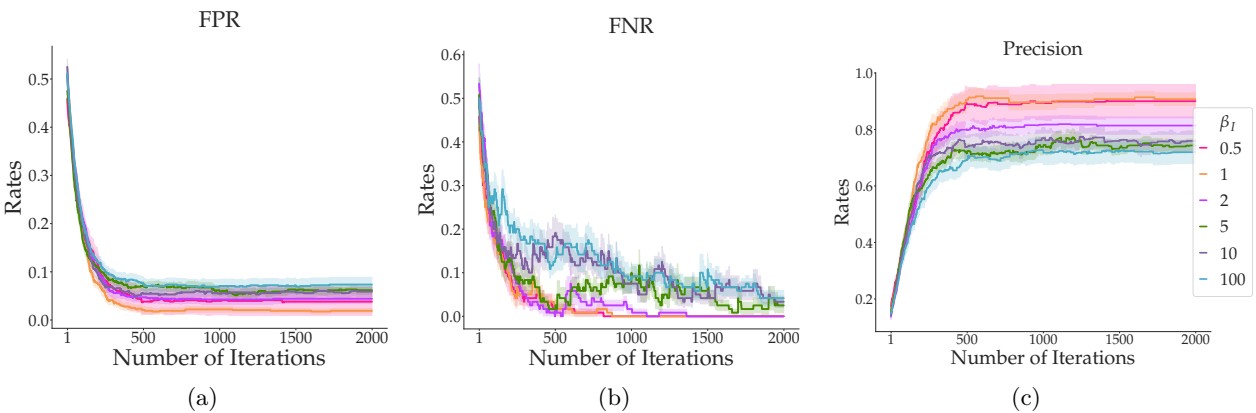

Figure 5: FPR (5a), FNR (5b) and Precision (5c) classification rates using BICRL in the grid world environment of Figure 1a under deterministic transition dynamics. Results averaged over 10 independent experiments.

Both the classification results, as well as the average constraint violation metric, showcase that a decomposition of inferring constraint indicator functions and the associated penalty reward leads to more accurate inference and safer policies. Constraint inference via BIRL leads to significantly higher constraint violation. BDPR and BCPR also underperform and some times lead to inferred penalty rewards that can alter the reward in a way that changes the actual goal state.

## 5.2 Demonstrator-Learner Discrepancy in Rationality

BICRL, as an inverse RL method, requires demonstrators from an expert in order to infer constraints. In general, it might be unrealistic to assume knowledge of the exact rationality levels of the demonstrator. In this section, we investigate the effect on inference when the rationality of the demonstrator captured, by the temperature $\beta_D$, is different from the rationality of the learner trying to infer the constraints, captured by $\beta_I$. For the environment and task depicted in Figure 1a, we gather 100 demonstrations from an expert with temperature parameter $\beta_D = 1$. We use BICRL to infer the constraints in that environment for a range of $\beta_I$ values.

We performed two sets of experiments, one assuming deterministic transition dynamics and another assuming stochastic dynamics with $\epsilon = 0.05$. We report the classification rates for each choice of $\beta_I$ in Figures 5 and 6. It should be noted that, as the value of $\beta$ increases the agent's behavior converges towards the optimal one. As expected the larger the discrepancy between $\beta_D$ and $\beta_I$ the higher the misclassification. Interestingly, this discrepancy is further attenuated by the stochasticity in the transition dynamics. One possible explanation for this phenomenon is the following. When $\beta_I$ has a high value, like 100, the inferring agent behaves almost optimally. However, the demonstrating agent is suboptimal and the additional noise exacerbates this. As a result, the inferring agent assumes that the demonstrating agent would rarely violate a constraint, and perhaps rarely move close to one, and hence ends up inferring significantly fewer constraints. To quantify the discrepancy between rationality levels more, in Section 8 we present inference results on a continuous state space environment obtained by using human demonstrations.

## 6 Hierarchical BICRL

For a large number of environments and tasks, safety constraints are compositional. In this section, we take advantage of that fact and we propose learning local constraints by observing agents perform sub-tasks. These local estimates can then be synthesized to obtain an estimate of the constraints in the entire space. Concretely, if the state space $S$ is made of distinct sub-domains $S_1, S_2, \ldots, S_n$, each with constraint sets $C_1, C_2, \ldots, C_n$, then the constraint set associated with the full state space is $C_1 \cup C_2 \cup \ldots \cup C_n$. By observing experts interact in those distinct domains, we can infer constraints that, when composed together, provide

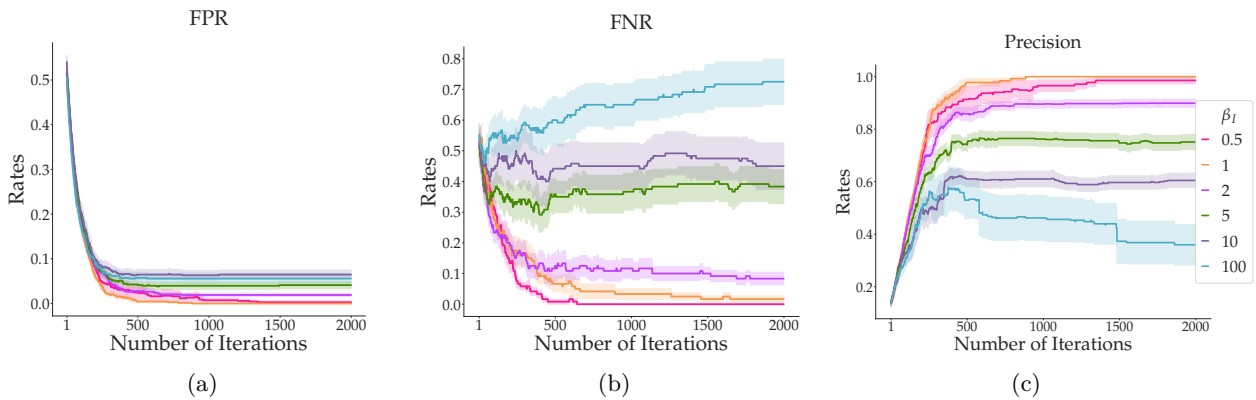

Figure 6: FPR (6a), FNR (6b) and Precision (6b) classification rates using BICRL in the grid world environment of Figure 1a under stochastic transition dynamics with noise level $\epsilon = 0.05$. Results averaged over 10 independent experiments.

more accurate constraint estimation as compared to a global inference approach, in which one agent tries to complete one task on $S$. Learning constraints from sub-tasks has also the benefit of mitigating the high dimensionality of the original problem. BICRL can be run in parallel for each sub-domain increasing the computational efficiency of our approach.

To showcase the benefits of learning from sub-tasks, we design an experiment on a $24 \times 24$ state grid world environment with constraints as seen in Figure 7a. We compare the case of *Global* inference in which the agent tries to traverse from the bottom right to the top left cell with the *Hierarchical* case in which the domain is split into four non-overlapping $12 \times 12$ sub-domains in which agents complete specific sub-tasks. Figure 7a shows the original grid world, along with the constraints and state visitation frequencies of the demonstrations, while Figure 8 shows the sub-domains extracted from the main environment and the associated sub-tasks.

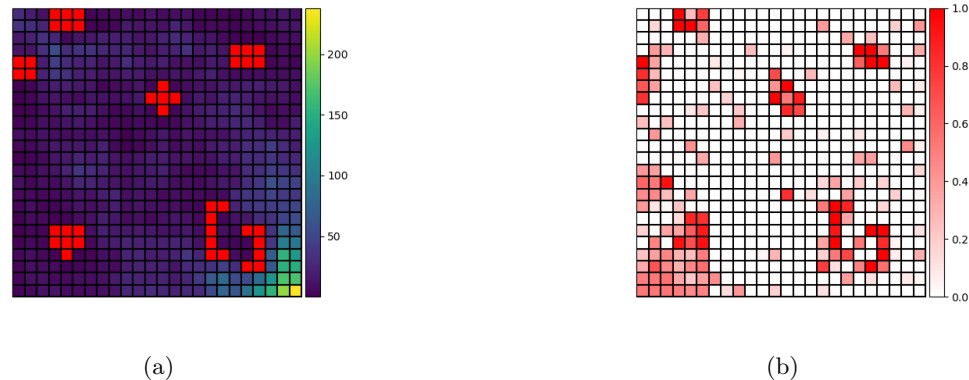

Figure 7: Even given 10000 samples, performing global constraint inference results in much more uncertainty than when using Hierarchical BICRL (Figures 8 and 9). (a) Original constraints along with state occupancies from 10000 samples from expert trajectories. (b) Global EAP constraint estimates.

In each of the four sub-domains, the experts have to complete a different task as shown by the blue (start state) and green (target state) flags respectively. The posterior mean estimates of the constraints are shown in Figure 9. In comparison, the posterior mean estimates of the *Global* inference case shown in Figure 7b are considerably more uncertain.

To further motivate *Hierarchical* BICRL, we compare the classification performance of the MAP estimates of the two methods. Given that trajectories, and hence the total number of state-action demonstration pairs, can

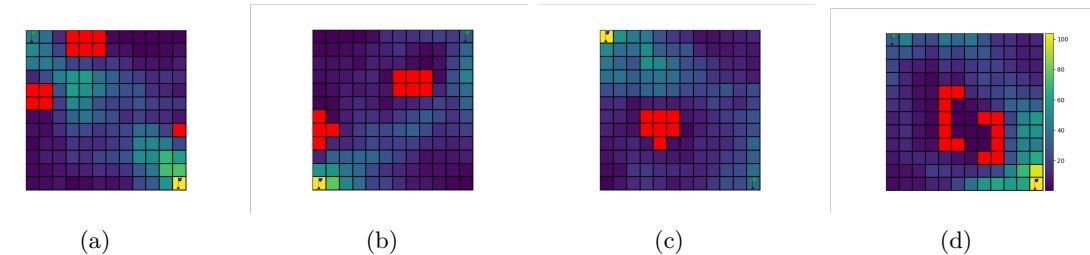

(a)          (b)          (c)          (d)

Figure 8: Original sub-task constraints and state occupancies for 2500 samples (for each sub-domain) from expert trajectories. The sub-tasks were obtained by splitting the original $24 \times 24$ grid into four equally sized quadrants.

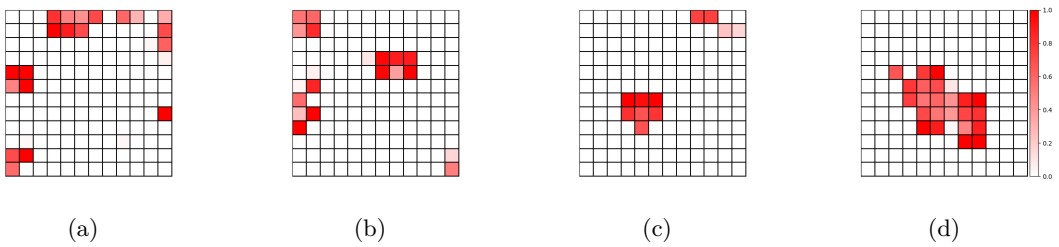

(a)          (b)          (c)          (d)

Figure 9: Hierarchical BICRL learns more accurate and less uncertain constraints than using global inference. Shown are the EAP constraint estimates for the four sub-domains of Figure 8.

be of varying length due to the grid size differences, for fairness we compare the two methods by varying the total number of state-action transition pairs used during inference. Table 2 clearly showcases that *Hierarchical* BICRL achieves significantly higher classification accuracy while providing a more computationally efficient alternative.

Table 2: False Positive, False Negative and Precision classification rates for *Global* and *Hierarchical* BICRL over a varying number of state-action demonstration sample sizes. Results averaged over 10 experiments.

| | BICRL (*Global*) | | | BICRL (*Hierarchical*) | | |
|---|---|---|---|---|---|---|
| Samples | FPR | FNR | Precision | FPR | FNR | Precision |
| 1000 | 0.34 | 0.38 | 0.12 | **0.26** | **0.35** | **0.15** |
| 5000 | 0.25 | **0.13** | 0.23 | **0.08** | 0.2 | **0.46** |
| 10000 | 0.12 | **0.08** | 0.43 | **0.03** | 0.11 | **0.67** |

The main goal of Hierarchical BICRL, except for the possible computational gains, is to show that constraint inference in the entire state space is still possible even if only sub-domain demonstrations are available. If only global task demonstrations are available, then Hierarchical BICRL would require further information about sub-tasks, which night not always be straightforward to obtain. It should be noted that, if diverse sub-task demonstrations are provided then classification might be more accurate as opposed to the case in which only demonstrations from a global task are available. That is expected, as higher diversity in the demonstrations leads to more accurate constraint inference due to better exploration of the state space.

## 6.1 Home Navigation Task

In this section, we present a more realistic constraint inference scenario for Hierarchical BICRL in which an agent infers obstacles in a home environment. An example of such an agent could be a home appliance

robot trying to map the rooms of a house. Figure 10 shows the floor plan of a single bedroom apartment obtained from the iGibson dataset (Li et al., 2021). A natural way to divide the entire apartment domain into sub-domains is to consider each room individually. The four room spaces are discretized and for each room we provide a number of expert demonstrations for two navigation sub-tasks. The details regarding the sub-tasks, associated rewards and the demonstrations can be seen in Appendix B.1.

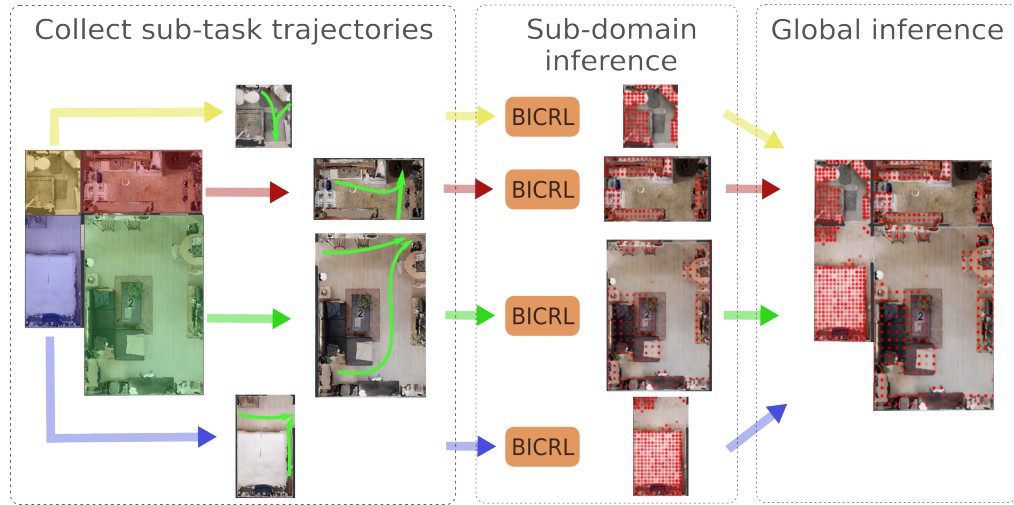

Figure 10: *Hierarchical* BICRL in a home navigation environment with posterior mean results.

In Figure 10, we show the mean constraint estimates of the posterior distribution for each sub-domain and ultimately for the entire space. BICRL manages to identify with high certainty most of the constraints in the rooms. As expected, sections of the domain with few or no demonstrations, like the bed area in the bottom left room, have higher uncertainty. Occasionally, false positives appear in the classification and they can be attributed to the stochastic nature of our Bayesian method. In practice, a large number of expert demonstrations that span the entire domain will drastically decrease the false positives in the classification. This example showcases, that given the decomposability of constraints, they can be efficiently inferred from independent sub-tasks. Having access to that information, an agent can then design policies and safely complete any task, possibly involving a subset of the sub-domains.

## 7    Feature-Based BICRL

This section introduces a feature-based version of BICRL. Estimating feature-based constraints allows for better generalization to different environments and tasks. In the feature-based case, the walk takes place on the feature weight vector and given that this is usually of smaller dimension than for instance the number of states in Section 5, BICRL may require fewer iterations for the posterior distribution to converge. Feature-based BICRL, detailed in Algorithm 5 in the Appendix, follows the same logic as the original version of the algorithm. The main difference is that this time the features and not the states are considered to be constrained or not. At each iteration, the algorithm either switches the label of a feature between constraint and free or samples a value for the feature weight associated with constraint features.

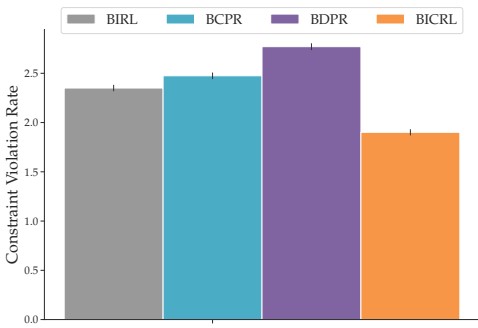

Figure 13: Average constraint violation in the test highway environment pictured in Figure 11b. Results averaged over 10 independent simulations.

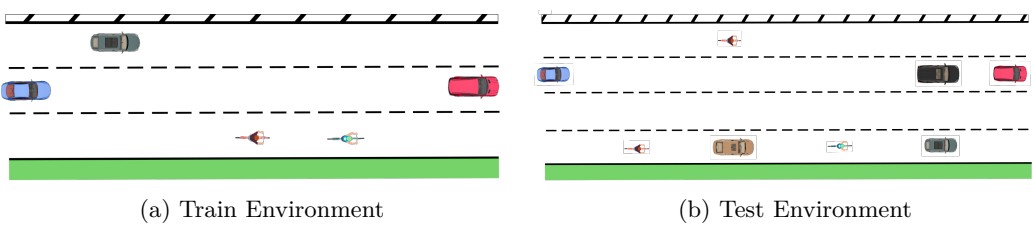

(a) Train Environment  (b) Test Environment

Figure 11: BICRL learns constraints from demonstrations in (a) that better transfer to new settings (b).

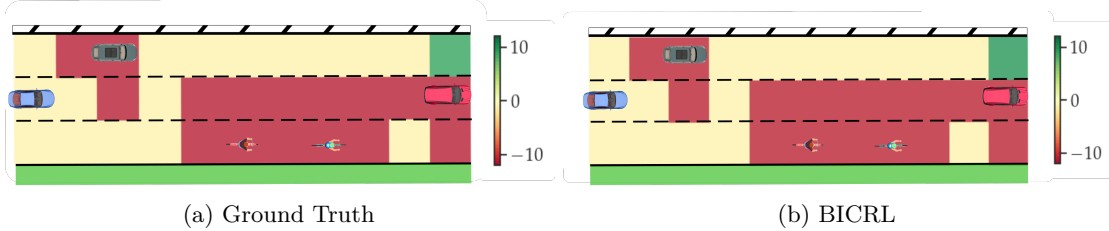

(a) Ground Truth  (b) BICRL

Figure 12: BICRL is able to learn the correct constraint function from driving demonstrations.

We study a highway environment in which the reward in each state $s$ is given as a linear function over features $R(s) = \mathbf{w}^T \phi(s)$. The nine binary features $\phi \in \mathbb{R}^9$ and the corresponding weights $\mathbf{w} \in \mathbb{R}^9$ are shown in Table 9 in the Appendix. The dimensions of the original highway environment shown in Figure 11a are $3 \times 11$. The goal of the driver of the blue car is to overtake the rest of the vehicles and the cyclists. For tailgating a car, overtaking from the right side of another car, passing close to the cyclists or any collision the driver incurs a penalty of $-10$. The driver obtains a reward of 10 for completing the task while being penalized for driving slowly. For 20 expert demonstrations and $K = 2000$ iterations, Figures 12a and 12b show the original reward and the one recovered from BICRL. The latter results are average over 10 independent simulations.

Finally, we compare how the inferred feature weights from BIRL, BCPR, BDPR and BICRL generalize to a new unseen environment shown in Figure 11b. The new environment models a longer highway stretch with more lanes and vehicles. For each method, we run 10 independent inference simulations using each time new expert demonstrations and for each of those we obtain 100 optimal trajectories using the inferred features. Figure 13 shows the average constraint violation in the test environment, providing evidence that BICRL leads to safer policies.

# 8 Continuous State Spaces

Finally, we investigate the performance of BICRL in continuous state spaces with human provided demonstrations. We utilize a two dimensional navigation task depicted in Figure 14a, in which the goal is to navigate from a starting state in the set $S_s$ to a state in $S_g$ while avoiding the constraint states in $S_c$. We collect a dataset of 20 human demonstrations by selecting way points from the start to the goal state that complete the task. The state space is comprised of $(x, y)$ coordinate tuples while the control inputs are the orientation $\phi \in [0, 2\pi]$ and the travel distance along the direction of the orientation $r \in [0, 0.25]$. After the way points have been created, the actions are computed by inverting the dynamics between each pair of adjacent way points.

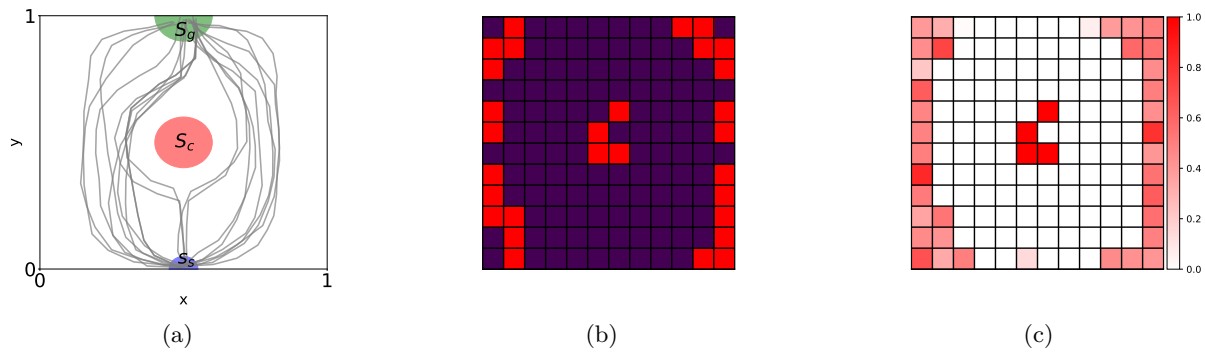

(a)                                    (b)                                    (c)

Figure 14: Continuous two dimensional navigation task along with human demonstrations (14a), MAP (14b) and EAP (14c) constraint estimates.

As BICRL applies to discrete state spaces, we proceed to discretize both the state and action spaces, in a similar way to Stocking et al. (2022). The state space is discretized into a $12 \times 12$ grid, while the action space is discretized into 8 actions for the orientation and 5 actions for the distance. We assume that the target states are associated with a reward of 5, while the living reward is $-2$. We fix the value of $\beta$ at 10, as the demonstrations are noisy but close to optimal. Figures 14b and 14c show the MAP and EAP estimates obtained from BICRL. While one of the constraint states is missclassified, the other three are identified correctly. As expected, in the unexplored areas of the state space uncertainty is high which is evident in the corresponding EAP estimates.

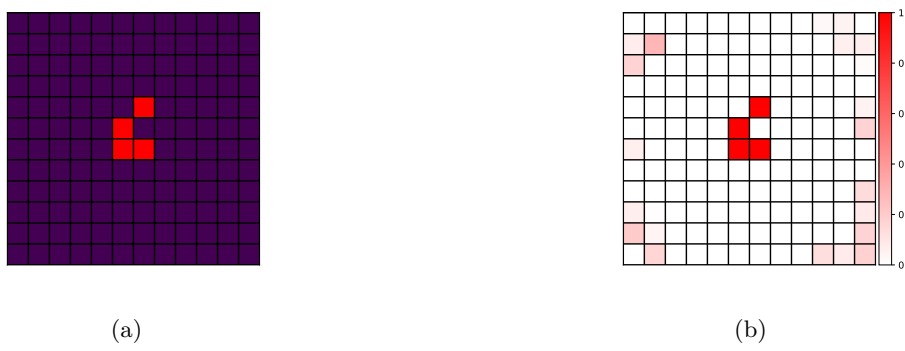

(a)                                    (b)

Figure 15: MAP (15a) and EAP (15b) constraint estimates under sparsity priors.

In the entirety of this paper, we assumed that there is no prior information regarding constraints. Given that in certain applications some information regarding constraints could be available, we also examine the classification performance of BICRL in the case where we assume that constraints are sparse. More

specifically, each state is assumed to have a prior probability of being a constraint that follows a Bernoulli distribution with parameter 0.05. For the same trajectories and hyperparameters as the ones used in the results in Figure 14, the MAP and EAP estimates for the sparse prior case are shown in Figure 15. The inclusion of a sparsity prior removes the noisy MAP estimates on the unexplored domain of the environment while keeping the constraint estimates around the actual constraint unaffected.

## 9    Conclusion

In this work we proposed BICRL, a Bayesian approach to infer the unknown constraints in discrete MDPs. BICRL can be utilized in both deterministic and stochastic environments as well as under complete or partial demonstrations. We showed that BICRL outperforms in constraint classification established approaches in both types of environments. The posterior distribution obtained allows for active learning tools to be utilized to further improve classification, especially in infrequently visited by the expert sections of the state space. We further proposed an hierarchical version of BICRL that allows for independent constraint inference in distinct sub-domains of the entire state space. Furthermore, we extended BICRL to feature-based environments and we showed that the estimated feature vectors can be used to obtain safe policies in new environments. Finally, we implemented BICRL in a continuous state space environment and we showed its effectiveness in inferring constraints from human demonstrations.

## 10    Limitations and Future Directions

Constraint learning is a relatively new research area, hence, there is significant room for improvement. In this work, we only consider constraint learning in low dimensional, and in most cases, discrete state spaces. An important avenue of future research, will be to adapt the principles introduced in this paper, to work with preference learning in order to enable scalable inference of constraints in high-dimensional continuous environments. Additionally, our active learning approach could be further improved through using more informative acquisition functions which could potentially incorporate information from the environment and corresponding task.

Another direction of future work is to leverage the fact that our constraint estimates are associated with confidence levels obtained from the posterior distribution. These confidence levels could be used to design policies that satisfy certain safety criteria, as agents can utilize this information to keep, for example, certain distance from areas where the existence of constraints is highly uncertain.

Hierarchical BICRL requires experts to provide demonstrations under various different reward functions. In many environments, such as the home navigation environment studied in this work, design of these reward functions may be straightforward; however, in other environments of interest, design of these different reward functions may be quite challenging. This may merit use of *Global* BICRL, over *Hierarchical* BICRL in environments where there is sufficiently diverse data available naturally. Finally, this work and other prior art in constraint learning has mainly focused on simulated studies. Application papers focused on utilizing constraint learning for real world applications is another exciting area of future research.

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

# A  Appendix

## A.1  Details for Simulations in Section 4

This section contains the details for the deterministic transition dynamics environment simulations, as well as the two stochastic scenaria of Section 4.2. The MDP has $8 \times 10$ number of states. The possible actions are *up*, *down*, *left* and *right*. The discount factor is $\gamma = 0.95$. Starting and goal states are depicted with a blue and green flag respectively. The expert trajectories are obtained with a Boltzmann policy model with $\beta = 1$. For this example, and all the rest of the simulations in this paper, we gather demonstrated trajectories by letting the expert agent follow a Boltzmann policy (7) from the start state until the goal state is reached or a predetermined number of steps has been made.

Table 3: Hyperparameters of Sections 4.1-4.3 simulations.

| Hyperparameters | Sec. 4.1 | Sec. 4.2 | Sec. 4.3 |
|---|---|---|---|
| # Expert trajectories | 100 | 100 | 20 |
| $n$ | 80 | 80 | 80 |
| $\gamma$ | 0.95 | 0.95 | 0.95 |
| $\epsilon$ | 0.0 | $0.0, 0.01, 0.05$ | 0.0 |
| $\beta$ | 1 | 1 | 1 |
| $K$ | 2000 | 4000 | 200 |
| $\sigma$ | 1 | 1 | 1 |
| $f_r$ | 50 | 50 | 50 |

In Section 4.2, we tune the KL divergence stopping criterion by using a grid search over its values. The classification results for these values for the deterministic and stochastic transition models can be seen in Table 4. The high levels of false negatives for $\epsilon > 0.0$ are attributed to the infrequent violation of constraints due to transition dynamics noise.

Table 4: False Positive, False Negative and Precision classification rates for GICI over varying KL divergence stopping criteria for environment in Figure 1a. Results averaged over 10 runs.

| KL | $\epsilon = 0.0$ | | | $\epsilon = 0.01$ | | | $\epsilon = 0.05$ | | |
|---|---|---|---|---|---|---|---|---|---|
| | FPR | FNR | Precision | FPR | FNR | Precision | FPR | FNR | Precision |
| 0.5 | 0.0 | 0.91 | 1.0 | 0.0 | 0.95 | 0.5 | 0.0 | 1.0 | 0.0 |
| $10^{-1}$ | 0.0 | 0.41 | 1.0 | 0.0 | 0.63 | 1.0 | 0.0 | 0.86 | 0.90 |
| $10^{-2}$ | 0.0 | 0.33 | 0.94 | 0.0 | 0.55 | 0.96 | 0.0 | 0.83 | 0.90 |
| $10^{-3}$ | 0.02 | 0.08 | 0.89 | 0.01 | 0.46 | 0.88 | 0.0 | 0.83 | 0.90 |
| $10^{-4}$ | 0.02 | 0.0 | 0.89 | 0.01 | 0.45 | 0.85 | 0.0 | 0.83 | 0.85 |

We also report the mean prediction for the constraints learned in Section 4.3. As expected, uncertainty is now significantly higher before running active learning. Applying Algorithm 2 from Section 3.5 with $K_Q = 20$, $K_A = 100$ and $K_D = 20$ we obtain mean estimates that are significantly more accurate as seen in Figure 16c. This feature of BICRL makes it appropriate for tasks where demonstrations are scarce and there is a limitation on the number of queries. In the active learning case, we first obtained initial constraint estimates by running BICRL on the initial 20 trajectories for 200 iterations and then used the active learning algorithm to further refine the classification.

## A.2  Details for Simulations in Section 5

This section contains the details on the simulations we run to compare our method with three other Bayesian approaches. The four MDP environments have $8 \times 10$ number of states. The possible actions are *up*, *down*,

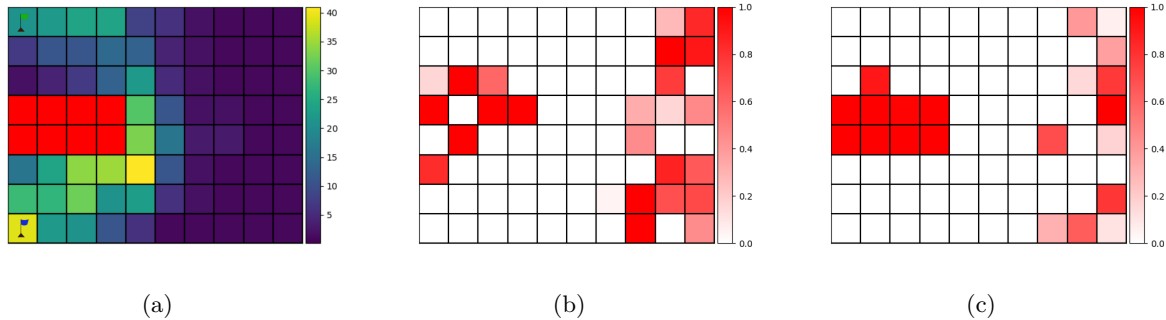

(a)                              (b)                              (c)

Figure 16: (16a) Original 20 trajectories and true constraint allocation. (16b) EAP constraint estimation without active learning querying. (16c) EAP constraint estimation after 10 active learning queries.

*left* and *right*. There is uncertainty in the transition model as with certain probability the agent moves to a neighboring state instead of the target state. For instance for noise level $\epsilon$ when the agent takes the action up then the agent will end up taking the action *up* with probability $1 - 2 * \epsilon$ and the actions *left* and *right* each with probability $\epsilon$. If the agent tries to transition outside the grid boundaries then the agent remains in the same state as shown in Figure 17.



Figure 17: Stochastic dynamics with $\epsilon = 0.05$ for taking action *up*. Red squares represent grid walls. The rest of the actions are analogous.

The discount factor is $\gamma = 0.95$. Starting and goal states are depicted with a blue and green flag respectively. For each of the MDPs we used 20 expert trajectories which we assume are noisy. We model this noise by using a Boltzmann policy (7) for the expert with $\beta = 1$.

Table 5: Hyperparameters of Section 5.1 simulations.

| Hyperparameters | Values |
|---|---|
| # Expert trajectories | 20 |
| $n$ | 80 |
| $\gamma$ | 0.95 |
| $\epsilon$ | 0.1 |
| $\beta$ | 1 |
| $K$ | 4000 |
| $\sigma$ | 1 |
| $f_r$ | 50 |

The algorithms for BDPR and BCPR are given below. We denote the reward penalty $\mathbf{r}_p \in \mathbb{R}^n$ with bold letter as now it is a vector since we infer a state-dependent penalty. Both BCPR and BDPR assume knowledge of the nominal reward and sample a state-dependent penalty reward that is added on top of the nominal reward. The MDP notation $M_{\mathbf{r}_p}$ refers to the MDP that has the penalty reward term added to its nominal reward function.

---

**Algorithm 3** BDPR

---

1: **Parameters:** Number of iterations $K$
2: Randomly sample reward vector $\mathbf{r}_p \in \mathbb{Z}^n$
3: $chain_{\mathbf{r}_p}[0] = \mathbf{r}_p$
4: Compute $Q^*$ on $M_{\mathbf{r}_p}$
5: **for** $i = 1, \ldots, K$ **do**
6:     Randomly sample state $j$ from $\{1, \ldots, K\}$
7:     Set $\mathbf{r}'_p[j] = \mathbf{r}_p[j] + 1$ or $\mathbf{r}_p[j] - 1$ with equal probability
8:     Compute $Q^*$ on $M_{\mathbf{r}'_p}$
9:     **if** $\log \mathcal{L}(\mathbf{r}'_p) \geq \log \mathcal{L}(\mathbf{r}_p)$ **then**
10:         Set $\mathbf{r}_p = \mathbf{r}'_p$
11:     **else**
12:         Set $\mathbf{r}_p = \mathbf{r}'_p$ w.p. $\mathcal{L}(\mathbf{r}'_p)/\mathcal{L}(\mathbf{r}_p)$
13:     $chain_{\mathbf{r}_p}[i] = \mathbf{r}_p$
14: **Return** $chain_{\mathbf{r}_p}$

---

**Algorithm 4** BCPR

---

1: **Parameters:** Number of iterations $K$, standard deviation $\sigma$
2: Randomly sample penalty reward vector $\mathbf{r}_p \in \mathbb{R}^n$
3: $chain_{\mathbf{r}_p}[0] = \mathbf{r}_p$
4: Compute $Q^*$ on $M_{\mathbf{r}_p}$
5: **for** $i = 1, \ldots, K$ **do**
6:     Randomly sample state $j$ from $\{1, \ldots, n\}$
7:     Set $\mathbf{r}'_p[j] \sim \mathcal{N}(\mathbf{r}_p[j], \sigma)$
8:     Compute $Q^*$ on $M_{\mathbf{r}'_p}$
9:     **if** $\log \mathcal{L}(\mathbf{r}'_p) \geq \log \mathcal{L}(\mathbf{r}_p)$ **then**
10:         Set $\mathbf{r}_p = \mathbf{r}'_p$
11:     **else**
12:         Set $\mathbf{r}_p = \mathbf{r}'_p$ w.p. $\mathcal{L}(\mathbf{r}'_p)/\mathcal{L}(\mathbf{r}_p)$
13:     $chain_{\mathbf{r}_p}[i] = \mathbf{r}_p$
14: **Return** $chain_{\mathbf{r}_p}$

---

Table 6 contains the hyperparameters regarding the sensitivity in the temperature parameter $(\beta)$ of Section 5.2.

Table 6: Hyperparameters of Section 5.2 simulations.

| Hyperparameters | Values |
|---|---|
| # Expert trajectories | 100 |
| $n$ | 80 |
| $\gamma$ | 0.95 |
| $\epsilon$ | $0.0, 0.05$ |
| $\beta_D$ | 1 |
| $\beta_I$ | $0.5, 1, 2, 5, 10, 100$ |
| $K$ | 2000 |
| $\sigma$ | 1 |
| $f_r$ | 50 |

## B    Hierarchical BICRL

For the Hierarchical BICRL in Section 6, we compare the methods using the varying number of state-action tuples obtained from the expert and not the number of trajectories. The reason for that, is that the heterogeneity in the constraints and grid sizes can lead to imbalanced numbers of total state-action demonstrations for the same number of expert trajectories. In practice, we keep gathering trajectories until the certain limit of state-action tuples is reached.

Table 7: Hyperparameters of grid world Hierarchical BICRL.

| Hyperparameters | Values |
|---|---|
| # Expert samples | $1000, 5000, 10000$ |
| $n$ (Global, Hierarchical) | $576, 4 \times 144$ |
| $\gamma$ | $0.95$ |
| $\epsilon$ | $0.1$ |
| $\beta$ | $1$ |
| $K$ | $4000$ |
| $\sigma$ | $1$ |
| $f_r$ | $50$ |

### B.1    Home Navigation with Hierarchical BICRL

For the home navigation task, we use one of the apartments in the iGibson dataset Li et al. (2021). Each room is discretized, counting from the top left clockwise, into $16 \times 15$, $12 \times 24$, $23 \times 17$ and $29 \times 15$ states respectively. In this set of experiments, the reward associated with the goal state is now 10 while the penalty reward and the living reward are $-10$ and $-1$ respectively.

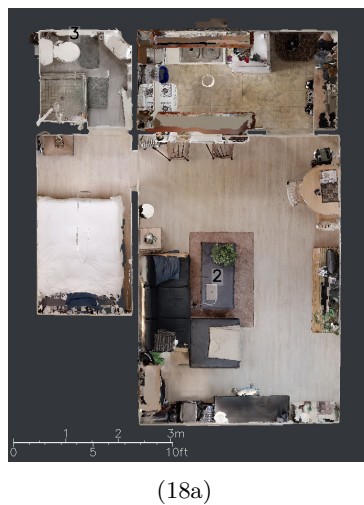
(18a)

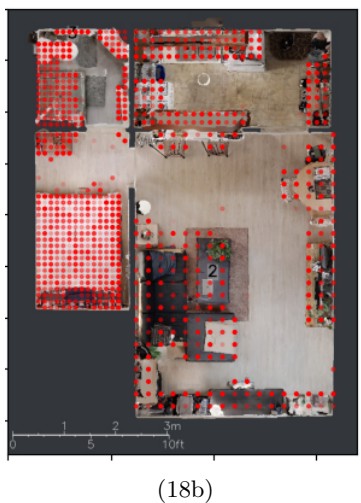
(18b)

Figure 18: Original four room layout (18a) and mean constraint estimates (18b) on the four rooms.

Figure 18 contains the original apartment layout and the posterior mean constraint estimates. The brightness of each grid point in the Figure 18 on the right is proportional to the likelihood of that state being constrained. Colorless grid points designate states that are estimated to be free of obstacles. The parameters used to infer the constraints in each room can be seen in Table 8.

Table 8: Hyperparameters of home navigation simulations.

| Hyperparameters | Values |
| --- | --- |
| # Expert trajectories | 100 |
| $n$ (for each room) | $240, 288, 391, 435$ |
| $\gamma$ | 0.95 |
| $\epsilon$ | 0.1 |
| $\beta$ | 1 |
| $K$ | 4000 |
| $\sigma$ | 1 |
| $f_r$ | 50 |

The expert trajectories gathered for the subtasks in the individual rooms can be seen in Figure 19. Each room has two individual subtasks that start from the blue flags and both end in the green flag. Figure 10, although more conceptual in nature, also shows the subtasks for each room with green arrows. For each of the rooms, we gather 50 expert trajectories for each subtask by using a Boltzmann policy with temperature parameter $\beta = 1$.

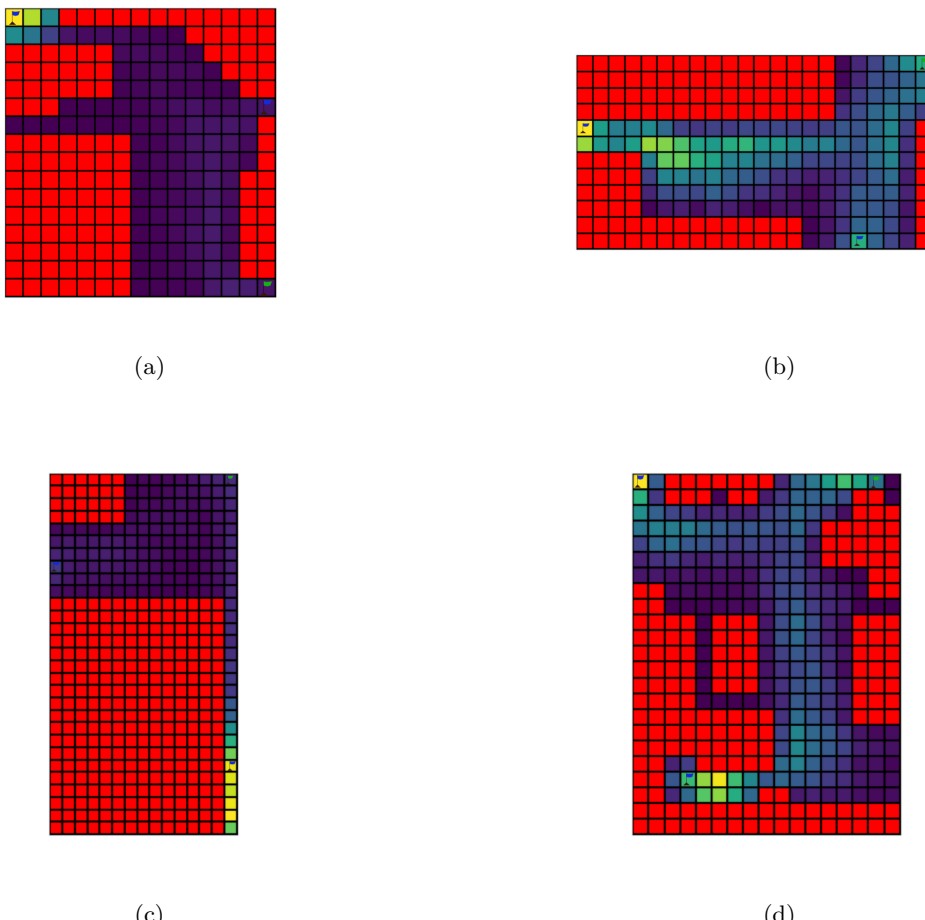

(a)                                                     (b)

(c)                                                     (d)

Figure 19: Home navigation expert trajectories. With reference to Figure 18a, top left (19a), top right (19b), bottom left (19c) and bottom right (19d) rooms.

# C   Feature-Based BICRL and Continuous State Space Navigation

## C.1   Feature-Based BICRL

Figure 20 shows the highway environment used for the feature-based simulations. The original environment is discretized in $3 \times 11$ states. The feature weights are shown in Table 9. The blue car pays a "living" reward penalty of $-1$, which penalizes driving slowly, while obtaining a small positive reward for overtaking a car from the left lane (car on the right). Overtaking a car from the right lane (car on the left), along with proximity to a cyclist, tailgating and collisions are considered constraints and are heavily penalized with a penalty reward of $-10$. The feature-based version of BICRL can be seen in Algorithm 5. BICRL, BCPR and BDPR assume knowledge of the "nominal" feature weights (features $4, 5, 6, 7$ and $9$), and they sample feature weights on top of these. In lines 5 an 12 in Algorithm 5, the Q values are computed on the MDP that has its nominal feature weights along with the inferred constraint ones that are associated with a penalty reward (weight) of $r_p$. For instance, if the current weight sample is $\mathbf{w}' = [0, 1, 0, 0, 0, 0, 0, 0, 0]$ and $r_p' = -2.5$ then the weight vector of $M_{\mathbf{w}', r_p'}$ is $\mathbf{w}' = [0, -2.5, 0, 1, -1, -1, -1, 0, 10]$. In this particular MDP, the goal state is reaching the top right cell by overtaking all the vehicles and cyclists according to the rules.

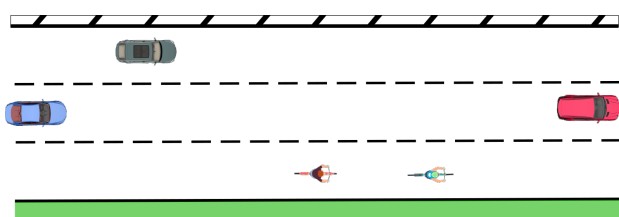

Figure 20: Highway environment.

The hyperparameters of the simulations can be seen in Table 10.

Table 9: Features and weights of highway environment.

| Feature ID | Feature | Weight |
|:---:|:---:|:---:|
| 1 | Tailgating | -10.0 |
| 2 | Close to cyclist | -10.0 |
| 3 | Car on the left | -10.0 |
| 4 | Car on the right | 1.0 |
| 5 | Driving on left lane | -1.0 |
| 6 | Driving on middle lane | -1.0 |
| 7 | Driving on right lane | -1.0 |
| 8 | Collision | -10.0 |
| 9 | Goal State | 10.0 |

Table 10: Hyperparameters of highway environment.

| Hyperparameters | Values |
|---|---|
| # Expert trajectories | 20 |
| $n$ | 33 |
| $n_\phi$ | 9 |
| $\gamma$ | 0.95 |
| $\epsilon$ | 0.1 |
| $\beta$ | 1 |
| $K$ | 2000 |
| $\sigma$ | 0.2 |
| $f_r$ | 20 |

---

**Algorithm 5** Feature-Based BICRL

1: **Parameters:** Number of iterations $K$, penalty reward sampling frequency $f_r$, standard deviation $\sigma$
2: Randomly sample $\mathbf{w} \in \mathbb{R}^{n_\phi}$
3: $chain_{\mathbf{w}}[0] = \mathbf{w}$
4: $chain_{r_p}[0] = r_p$
5: Compute $Q^*$ on $M_{\mathbf{w}, r_p}$
6: **for** $i = 1, \ldots, K$ **do**
7:     **if** $(i \bmod f_r)$!=0 **then**
8:         Randomly sample feature $j$ from $\{1, \ldots, n_\phi\}$
9:         Set $\mathbf{w}'[j] = \neg\mathbf{w}[j]$, $r_p' = r_p$
10:     **else**
11:         Set $r_p' = r_p + \mathcal{N}(0, \sigma)$, $\mathbf{w}' = \mathbf{w}$
12:     Compute $Q^*$ on $M_{\mathbf{w}', r_p'}$
13:     **if** $\log \mathcal{L}(\mathbf{w}', r_p') \geq \log \mathcal{L}(\mathbf{w}, r_p)$ **then**
14:         Set $\mathbf{w} = \mathbf{w}'$, $r_p = r_p'$
15:     **else**
16:         Set $\mathbf{w} = \mathbf{w}'$, $r_p = r_p'$ w.p. $\mathcal{L}(\mathbf{w}')/\mathcal{L}(\mathbf{w})$
17:     $chain_{\mathbf{w}}[i] = \mathbf{w}$
18:     $chain_{r_p}[i] = r_p$
19: **Return** $chain_{\mathbf{w}}$, $chain_{r_p}$

---

### C.2 Continuous State Space Navigation

The hyperparameters used in Section 8 can be seen in table 11.

Table 11: Hyperparameters of continuous state space navigation environment.

| Hyperparameters | Values |
|---|---|
| # Expert trajectories | 20 |
| $n$ | 144 |
| # of actions | $8 \times 5$ |
| $\gamma$ | 0.95 |
| $\epsilon$ | 0.05 |
| $\beta$ | 10 |
| $K$ | 6000 |
| $\sigma$ | 1 |
| $f_r$ | 50 |

To collect the trajectories, a graphical user interface is used on which way points are created on the environment. The continuous environment state space is a $[0,1] \times [0,1]$ box which is then discretized into a $12 \times 12$ grid. Each continuous state is then assigned to each closest discrete state following a nearest neighbor rule. The discretization of actions follows a similar rule. The original continuous actions along a trajectory are obtained by inverting the dynamics between two adjacent way points.

## D   Theoretical Analysis and Proof of Rapid MCMC Mixing

### D.1   Preliminaries

Below, we provide a rigorous proof of rapid mixing for the Policy Walk algorithm found in Ramachandran & Amir (2007a). Note that while our proof follows theirs, there are some important differences. First, we note that the likelihood for Bayesian IRL (and the one used in our paper for BICRL) is of the form

$$P(D|R) = \prod_{(s,a) \in D} \frac{e^{\beta Q_R^*(s,a)}}{\sum_{b \in A} e^{\beta Q_R^*(s,b)}}. \tag{8}$$

Note that Ramachandran & Amir (2007a) assume the following likelihood function

$$P(D|R) = \prod_{(s,a) \in D} e^{\beta Q_R^*(s,a)}. \tag{9}$$

To see why this might not be accurate, consider Equation (3) in Ramachandran & Amir (2007a). The posterior probability of a reward function R by applying Bayes' theorem is given as

$$P(R|D) = \frac{P(D|R)P(R)}{P(D)} \tag{10}$$

$$= \frac{1}{Z'} e^{\beta \sum_{(s,a) \in D} Q_R^*(s,a)} P(R). \tag{11}$$

Ramachandran & Amir (2007a) then claim that computing the normalization $Z'$ is hard, which is true, but that the term can be ignored since MCMC only requires sampling probability density ratios. However, $Z'$ does not actually cancel when performing probability density ratios since $Z'$ is dependent on $R$. To see this we unpack the above posterior distribution

$$P(R|D) = \frac{P(D|R)P(R)}{P(D)} \tag{12}$$

$$= \frac{\prod_{(s,a) \in D} \frac{e^{\beta Q_R^*(s,a)}}{\sum_{b \in A} e^{\beta Q_R^*(s,b)}} P(R)}{P(D)} \tag{13}$$

$$= \frac{e^{\beta \sum_i Q_R^*(s_i,a_i)} P(R)}{\prod_{(s,a)} \sum_{b \in A} e^{\beta Q_R^*(s,b)} P(D)} P(R). \tag{14}$$

Thus, there is a clear dependence of $Z'$ on $R$ and when we take a ratio of probability densities for the posterior probability of two reward functions $R$ and $R'$, we cannot fully cancel $Z'$ as claimed by Ramachandran & Amir (2007a). Instead we have

$$\frac{P(R|D)}{P(R'|D)} = \frac{\frac{\exp(\beta \sum_i Q_R^*(s_i,a_i))P(R)}{\prod_{(s,a)} \sum_{b \in A} \exp(\beta Q_R^*(s,b))P(D)} P(R)}{\frac{\exp(\beta \sum_i Q_{R'}^*(s_i,a_i))P(R')}{\prod_{(s,a)} \sum_{b \in A} \exp(\beta Q_{R'}^*(s,b))P(D)} P(R')} \tag{15}$$

$$= \frac{\frac{\exp(\beta \sum_i Q_R^*(s_i,a_i))P(R)}{\prod_{(s,a)} \sum_{b \in A} \exp(\beta Q_R^*(s,b))} P(R)}{\frac{\exp(\beta \sum_i Q_{R'}^*(s_i,a_i))P(R')}{\prod_{(s,a)} \sum_{b \in A} \exp(\beta Q_{R'}^*(s,b))} P(R')}. \tag{16}$$

Thus, while $P(D)$ cancels, the rest does not. The reason this is important is because the proof of rapid mixing in Ramachandran & Amir (2007a) assumes that the log of the posterior probability for a reward sample in MCMC is

$$f(R) = \beta \sum_{(s,a) \in D} Q_R^*(s, a). \tag{17}$$

This makes the proofs incorrect, since the counterfactual Q-value terms in the likelihood function are ignored, namely the Q-values of alternative actions $b$.

It is easy to show that using the likelihood function presented in the proofs of rapid mixing by Ramachandran & Amir (2007a) can lead to a trivial and non-sensical reward. For simplicity we assume a uniform prior so we can focus on the likelihood function. If we choose the likelihood function to be

$$P(D|R) = \exp(\beta \sum_{(s,a) \in D} Q_R^*(s, a)), \tag{18}$$

then setting $R(s) = R_{\max}, \forall s$ maximizes the likelihood function; however, as noted by Ng et al. (2000), a constant reward function makes every policy optimal, and gives no insight into the reward function of optimal policy of the demonstrator. Thus, if we want to find a reward function that, when optimized, leads to behavior similar to the demonstrator, we need a likelihood function that maximizes the probability of the demonstrations. This calculation, by necessity, must then include the Q-values of the actions not taken by the demonstrator, leading us to the likelihood function used in our paper and shown in Equation (8). In the next section, we first seek to remedy this flaw in the proof of rapid mixing for Bayesian IRL. Then in the following section we show that this analysis also applies to BICRL.

### D.2 Rapid Mixing Theorem for Bayesian IRL

We make use of the following Lemma from Applegate & Kannan (1991) and Ramachandran & Amir (2007a).

**Lemma D.1.** *Let $F(\cdot)$ be a positive real valued function defined on the cube $\{x \in \mathbb{R}^n : -d \leq x_i \leq d\}$ and $f(x) = \log F(x)$. If there exist real numbers $L$ and $\kappa$ such that*

$$|f(x) - f(y)| \leq L\|x - y\|_\infty, \tag{19}$$

*and*

$$f(\lambda x + (1 - \lambda)y) \geq \lambda f(x) + (1 - \lambda)f(y) - \kappa, \tag{20}$$

*for all $\lambda \in [0, 1]$, then the Markov chain induced by GridWalk and PolicyWalk on $F$ rapidly mixes to within $\epsilon$ of $R$ in $O(n^2 d^2 L^2 e^{2\kappa} \log \frac{1}{\epsilon})$ steps.*

*Proof.* See Applegate & Kannan (1991). □

We will also need the following Lemma relating the Lipschitz-smoothness of value functions and reward functions. Here, we use the vectorized notation of value functions and reward functions for notational simplicity, but note that the same analysis applies to continuous MDPs.

**Lemma D.2.**

$$|V_{R_1}^*(s) - V_{R_2}^*(s)| \leq \frac{1}{1 - \gamma}\|R_1 - R_2\|_\infty \tag{21}$$

*Proof.*

$$\|V_{R_1}^* - V_{R_2}^*\|_\infty \leq \|R_1 + \gamma P_{\pi_1^*} V_{R_1}^* - R_2 - \gamma P_{\pi_2^*} V_{R_2}^*\|_\infty \tag{22}$$

$$\leq \|R_1 - R_2\|_\infty + \gamma\|P_{\pi_1^*} V_{R_1}^* - \gamma P_{\pi_2^*} V_{R_2}^*\|_\infty \tag{23}$$

$$\leq \|R_1 - R_2\|_\infty + \gamma\|P_{\pi_1^*}(V_{R_1}^* - V_{R_2}^*)\|_\infty \tag{24}$$

$$\leq \|R_1 - R_2\|_\infty + \gamma\|P_{\pi_1^*}\|_\infty\|V_{R_1}^* - V_{R_2}^*\|_\infty \tag{25}$$

$$\leq \|R_1 - R_2\|_\infty + \gamma\|V^*_{R_1} - V^*_{R_2}\|_\infty. \tag{26}$$

Thus, we have

$$\|V^*_{R_1}(s,a) - V^*_{R_2}(s,a)\|_\infty \leq \frac{1}{1-\gamma}\|R_1 - R_2\|_\infty, \tag{27}$$

and this gives us the final desired result

$$|V^*_{R_1}(s) - V^*_{R_2}(s)| \leq \max_s |V^*_{R_1}(s) - V^*_{R_2}(s)| = \|V^*_{R_1}(s,a) - V^*_{R_2}(s,a)\|_\infty \leq \frac{1}{1-\gamma}\|R_1 - R_2\|_\infty. \tag{28}$$

$\square$

**Theorem D.3.** *Given an MDP, $M = (S, A, T, \gamma)$ with $|S| = N$, and a distribution over rewards $P(R) = P(R|D)$ defined by Equation (14) with uniform prior $P(R)$ over the cube $C = \{R \in \mathbb{R}^n : -R_{\max} \leq R_i \leq R_{\max}\}$, if $R_{\max} = O(1/N)$ and $|A| = O(1)$, then $P(R)$ can be efficiently sampled within error $\epsilon$ in $O(N^2 \log 1/\epsilon)$ steps by Bayesian IRL (Ramachandran & Amir, 2007a).*

As stated in the theorem, we assume a uniform prior and that $|A| = O(1)$. Note that it is common that the number of actions is constant so this is not constraining. We will also assume that $R_{\max} = O(1/N)$. As noted by Ramachandran & Amir (2007a), this is not restrictive since we can rescale the rewards (and hence the value functions and Q-value functions) by a constant factor $k$ after computing the posterior without changing the optimal policy.

To show rapid mixing, we need to prove that there exist $L$ and $\kappa$ as prescribed in Lemma D.1. Assuming a uniform prior, we can ignore the prior. We now focus on the likelihood function for BICRL. The likelihood function is

$$P(D|R) = \prod_{(s,a) \in D} \frac{e^{\beta Q^*_R(s,a)}}{\sum_{b \in A} e^{\beta Q^*_R(s,b)}}. \tag{29}$$

Thus, we let $F(R) = \prod_{(s,a) \in D} \frac{e^{\beta Q^*_R(s,a)}}{\sum_{b \in A} e^{\beta Q^*_R(s,b)}}$ and $f(R) = \log F(R) = \beta \sum_i Q^*_R(s_i, a_i) - \sum_i \log \sum_{b \in A} e^{\beta Q^*_R(s_i, b)}$. We first consider the Lipschitz property. We have

$$\left| \beta Q^*_{R_1}(s,a) - \log \sum_{b \in A} e^{\beta Q^*_{R_1}(s,b)} - \beta Q^*_{R_2}(s,a) + \log \sum_{b \in A} e^{\beta Q^*_{R_2}(s,b)} \right| \leq \tag{30}$$

$$\beta \left| Q^*_{R_1}(s,a) - Q^*_{R_2}(s,a) \right| + \left| \log \sum_{b \in A} e^{\beta Q^*_{R_2}(s,b)} - \log \sum_{b \in A} e^{\beta Q^*_{R_1}(s,b)} \right|. \tag{31}$$

We now consider each term individually. Starting with the first term, we have

$$\beta \left| Q^*_{R_1}(s,a) - Q^*_{R_2}(s,a) \right| \leq \frac{2\beta}{1-\gamma}\|R_1 - R_2\|_\infty, \tag{32}$$

by Lemma 1 in the Appendix of Barreto et al. (2017). We now consider the second term. Recall that

$$\max\{x_1, \dots, x_n\} \leq \log \sum_i \exp x_i \leq \max\{x_1, \dots, x_n\} + \log n. \tag{33}$$

We thus have

$$\left| \log \sum_{b \in A} e^{\beta Q^*_{R_2}(s,b)} - \log \sum_{b \in A} e^{\beta Q^*_{R_1}(s,b)} \right| \leq \left| \max_b \beta Q^*_{R_2}(s,b) + \log |A| - \max_b \beta Q^*_{R_1}(s,b) \right| \tag{34}$$

$$= \left| \beta V^*_{R_2}(s) + \log |A| - \beta V^*_{R_1}(s) \right| \tag{35}$$

$$\leq \log |A| + \beta \left| V^*_{R_2}(s) - V^*_{R_1}(s) \right| \tag{36}$$

$$\leq \log |A| + \frac{\beta}{1 - \gamma} \|R_1 - R_2\|_\infty \tag{37}$$

$$= O(\frac{\beta}{1 - \gamma} \|R_1 - R_2\|_\infty), \tag{38}$$

where the second to last line follows from Lemma D.2 and the last line follows from our assumption that $|A| = O(1)$. The above analysis shows that we have

$$|f(R_1) - f(R_2)| = \left| \beta \sum_i Q_{R_1}^*(s_i, a_i) - \sum_i \log \sum_{b \in A} e^{\beta Q_{R_1}^*(s_i, b)} - \beta \sum_i Q_{R_2}^*(s_i, a_i) + \sum_i \log \sum_{b \in A} e^{\beta Q_{R_2}^*(s_i, b)} \right| \tag{39}$$

$$\leq \beta \sum_i \left| Q_{R_1}^*(s_i, a_i) - \beta Q_{R_2}^*(s_i, a_i) \right| + \sum_i \left| \log \sum_{b \in A} e^{\beta Q_{R_2}^*(s_i, b)} - \log \sum_{b \in A} e^{\beta Q_{R_1}^*(s_i, b)} \right| \tag{40}$$

$$\leq \frac{2N\beta}{1 - \gamma} \|R_1 - R_2\|_\infty + \frac{N\beta}{1 - \gamma} \|R_1 - R_2\|_\infty \tag{41}$$

$$= \frac{3N\beta}{1 - \gamma} \|R_1 - R_2\|_\infty. \tag{42}$$

We now turn to showing approximate log-concavity. For any arbitrary policy $\pi$ let

$$f_\pi(R) = \beta \sum_i Q_R^\pi(s_i, a_i, R) - \sum_i \log \sum_b e^{\beta Q_R^\pi(s, b)}. \tag{43}$$

We have that

$$f(R) = \beta \sum_i Q_R^*(s_i, a_i) - \sum_i \log \sum_b e^{\beta Q_R^*(s_i, b)} \tag{44}$$

$$\leq \beta \sum_i Q_R^*(s_i, a_i) - \sum_i \max_b \beta Q_R^*(s_i, b) \tag{45}$$

$$\leq \beta \sum_i Q_R^*(s_i, a_i) - \sum_i \beta V_R^*(s_i) \tag{46}$$

$$\leq 0, \tag{47}$$

where we have again used the fact that

$$\max\{x_1, \ldots, x_n\} \leq \log \sum_i \exp x_i \leq \max\{x_1, \ldots, x_n\} + \log n, \tag{48}$$

and also that $V^*(s) = \max_a Q^*(s, a)$. Thus, we also have

$$f_\pi(R) = \beta \sum_i Q^\pi(s_i, a_i, R) - \sum_i \log \sum_b e^{\beta Q_R^\pi(s, b)} \tag{49}$$

$$\geq -\beta \sum_i \frac{R_{\max}}{1 - \gamma} - \sum_i \log \sum_b e^{\beta Q_R^\pi(s, b)} \tag{50}$$

$$\geq -\beta \sum_i \frac{R_{\max}}{1 - \gamma} - \sum_i (\frac{\beta R_{\max}}{1 - \gamma} + \log |A|) \tag{51}$$

$$\geq -\frac{\beta R_{\max} N}{1 - \gamma} - N(\frac{\beta R_{\max}}{1 - \gamma} + \log O(1)) \tag{52}$$

$$\geq -\frac{2\beta R_{\max} N}{1 - \gamma} \tag{53}$$

$$\geq f(R) - \frac{2\beta R_{\max} N}{1 - \gamma}, \tag{54}$$

where we have used the fact that Q-values are upper bounded by $R_{\max}/(1-\gamma)$ and where the last line comes from the fact that $f(R)$ is non-positive.

We can now prove approximate log-concavity. Let $R' = \lambda R_1 + (1-\lambda)R_2$, then we have

$$f(\lambda R_1 + (1-\lambda)R_2) = \beta \sum_i Q_{R'}^{\pi_{R'}^*}(s_i, a_i) - \sum_i \log \sum_{b \in A} e^{\beta Q_{R'}^{\pi_{R'}^*}(s_i, b)} \tag{55}$$

$$= \beta \sum_i \mathbb{E}_{\pi_{R'}^*}\left[\sum_{t=0}^{\infty} \gamma^t R'(s_t, a_t)|s_0 = s_i, a_0 = a_i\right] \tag{56}$$

$$- \sum_i \log \sum_{b \in A} e^{\beta \mathbb{E}_{\pi_{R'}^*}\left[\sum_{t=0}^{\infty} \gamma^t R'(s_t, a_t)|s_0 = s_i, a_0 = a_i\right]} \tag{57}$$

$$= \beta \sum_i \mathbb{E}_{\pi_{R'}^*}\left[\sum_{t=0}^{\infty} \gamma^t \lambda R_1(s_t, a_t) + \gamma^t(1-\lambda)R_2(s_t, a_t)|s_0 = s_i, a_0 = a_i\right] \tag{58}$$

$$- \sum_i \log \sum_{b \in A} e^{\beta \mathbb{E}_{\pi_{R'}^*}\left[\sum_{t=0}^{\infty} \gamma^t \lambda R_1(s_t, a_t) + \gamma^t(1-\lambda)R_2(s_t, a_t)|s_0 = s_i, a_0 = a_i\right]} \tag{59}$$

$$= \beta \sum_i \mathbb{E}_{\pi_{R'}^*}\left[\sum_{t=0}^{\infty} \gamma^t \lambda R_1(s_t, a_t)|s_0 = s_i, a_0 = a_i\right] + \gamma^t(1-\lambda)\mathbb{E}_{\pi_{R'}^*}\left[R_2(s_t, a_t)|s_0 = s_i, a_0 = a_i\right] \tag{60}$$

$$- \sum_i \log \sum_{b \in A} e^{\beta \mathbb{E}_{\pi_{R'}^*}\left[\sum_{t=0}^{\infty} \gamma^t \lambda R_1(s_t, a_t)|s_0 = s_i, a_0 = a_i\right] + \gamma^t(1-\lambda)\mathbb{E}_{\pi_{R'}^*}\left[R_2(s_t, a_t)|s_0 = s_i, a_0 = a_i\right]} \tag{61}$$

$$= \beta \sum_i \lambda Q_{R_1}^{\pi_{R'}^*}(s_i, a_i) + (1-\lambda)Q_{R_2}^{\pi_{R'}^*}(s_i, a_i) \tag{62}$$

$$- \sum_i \log \sum_{b \in A} e^{\beta \lambda Q_{R_1}^{\pi_{R'}^*}(s_i, b) + (1-\lambda)Q_{R_2}^{\pi_{R'}^*}(s_i, b)} \tag{63}$$

$$\geq \beta \sum_i \lambda Q_{R_1}^{\pi_{R'}^*}(s_i, a_i) + \beta \sum_i (1-\lambda)Q_{R_2}^{\pi_{R'}^*}(s_i, a_i) \tag{64}$$

$$- \sum_i \lambda \log \sum_{b \in A} e^{\beta Q_{R_1}^{\pi_{R'}^*}(s_i, b)} - \sum_i (1-\lambda)\log \sum_b e^{Q_{R_2}^{\pi_{R'}^*}(s_i, b)} \tag{65}$$

$$= \lambda f_{\pi_{R'}^*}(R_1) + (1-\lambda)f_{\pi_{R'}^*}(R_2) \tag{66}$$

$$\geq \lambda\left(f(R_1) - \frac{2\beta R_{\max}N}{1-\gamma}\right) + (1-\lambda)\left(f(R_2) - \frac{2\beta R_{\max}N}{1-\gamma}\right) \tag{67}$$

$$= \lambda f(R_1) + (1-\lambda)f(R_2) - \frac{2\beta R_{\max}N}{1-\gamma}, \tag{68}$$

where Line (65) follows from the convexity of the log-sum-exponential and Line (67) follows from Equation (54). Thus, we have

$$L = \frac{3N\beta}{1-\gamma}, \tag{69}$$

and

$$\kappa = \frac{2\beta R_{\max}N}{1-\gamma}. \tag{70}$$

Thus, by Lemma D.1, the Markov chain induced by Bayesian IRL mixes rapidly to within $\epsilon$ of $P$ in a number of steps equal to $O(N^2 R_{\max}^2 L^2 \exp(2\kappa) \log \frac{1}{\epsilon}) = O\left(N^2 \frac{1}{N^2}\left(\frac{3N\beta}{1-\gamma}\right)^2 \exp\left(\frac{2\beta \frac{1}{N}N}{1-\gamma}\right)\log \frac{1}{\epsilon}\right) = O(N^2 \log 1/\epsilon).$

### D.3 Extension to BICRL

We now show that the above proof also extends to the Bayesian Inverse Constrained RL (BICRL) algorithm that we have proposed in this paper.

**Corollary D.3.1.** *BICRL has the same rapid mixing properties as Bayesian IRL.*

*Proof.* Note first that in BICRL we sample over the binary hypercube $\{0,1\}^N$ where $N$ is the number of states. We also sample over the constraint penalty scalar $r_p$. In practice we have $-r_p$ in the range $[0, R_{\max}]$, so BICRL samples are just a special case of sampling over the cuber $\{x \in \mathbb{R}^{n+1} : 0 \leq x_i \leq R_{\max}\}$. BICRL assumes access to a known task reward and then computes a posterior probability using Q-values over the augmented reward function given by

$$R^c(s,a) = \begin{cases} r_p & \text{if} \quad \mathbb{I}_C(s,a) \text{ is } 1 \\ R(s,a) & \text{otherwise.} \end{cases} \tag{71}$$

Thus, BICRL satisfies the same conditions as Bayesian IRL and by Lemma D.1, the Markov chain induced by BICRL mixes rapidly to within $\epsilon$ of $P$ in a number of steps equal to $O(N^2 \log 1/\epsilon)$. $\quad\square$

