# OpenReview forum: "Bayesian Methods for Constraint Inference in Reinforcement Learning"
_TMLR — Accepted by TMLR_

### Review · Reviewer_8ZEH · 2022-07-21

**Summary Of Contributions:**

In this paper, the authors study the problem of inferring constraints from demonstrations in a reinforcement learning setting. To this end, they propose Bayesian Inverse Constrained Reinforcement Learning (BICRL), a method that acquires the posterior probability over constraints given expert demonstrations. Extensive numerical evaluations are provided to support the method.

**Broader Impact Concerns:**

None.

**Requested Changes:**

Addressing the previously mentioned weaknesses should be sufficient.

**Strengths And Weaknesses:**

Strengths.

This is a work of possible interest to the community. Writing and presentation are clear and, for the most part, concepts and statements are clearly introduced. The main idea introduced in the paper, Bayesian Inverse Constrained Reinforcement Learning, is simple yet appears to work well in practice. This behavior is supported by extensive numerical evaluations.

Weaknesses.

- It feels like the paper is missing some more theoretical support for the proposed algorithm. It would increase the value of the manuscript to provide a more thorough analysis of the methodology leading to the algorithm and to evaluate some of its properties. As it currently stands, the manuscript is just a very simple idea applied to a large number of numerical scenarios.

- Isn't setting =0 in the constraint in (1)-(2) very restrictive? This is asking to satisfy the constraint with probability 1 (zero constraint violation), which is unfeasible in practice for most stochastic policies. Would it not be better to allow for some epsilon-probability (in (1), a constant epsilon instead of 0)?

- In section 4.1, "As expected, the agent demonstrates high uncertainty in areas that are far away from the expert demonstrations, like the bottom right section of the grid." Out of curiosity, could you go into more detail about why this behavior occurs (e.g., only on the bottom-right and not on the top-right corner)?

- While numerical extensive, they are all in discrete or grid world domains. It would be very interesting to evaluate the proposed methodology in continuous spaces rather than only grid world environments.

---

> ### Author Response · Authors · 2022-08-26
> **Response to Reviewer 8ZEH**
>
> We would first like to thank the reviewer for all the insightful comments. Below, we address each requested change individually.
>
> **It feels like the paper is missing some ...**
>
> We appreciate the suggestion. We have added a proof of rapid MCMC mixing to Appendix D in the updated submission. Our proof is inspired by the proof in Ramachandran and Amir’s original Bayesian IRL proof. However, while working through their proof we found that their proof relied on simplifying their likelihood function by ignoring the denominator in the Boltzmann equation. As we discuss in the appendix, this does not match the likelihood actually used in their paper and can lead to trivial reward functions having maximum likelihood. Thus, our theoretical contributions that we have added are two-fold. First, we extend the proof of Ramachandran and Amir to the full Bolztmann likelihood function and show that we can still obtain a proof of rapid mixing while using the correct likelihood function. Second, we prove that BICRL inherits the same rapid mixing properties as Bayesian IRL. We believe this new theory is interesting in its own right, as it provides more rigorous analysis of rapid mixing properties of the original Bayesian IRL algorithm. We also thank the author for their suggestion as it strengthens our paper by showing that BICRL also has rapid mixing theoretical guarantees.
>
>
> **Isn't setting =0 in ...**
>
> Assumption of epsilon (probability of violating a constraint)=0 is borrowed from prior literature (Anwar et al. and Scobee & Sastry) and is grounded in the fact that if it is not 0, then constraints no longer remain Markovian (see See footnote on page 2 for definition of Markovian constraints: https://arxiv.org/pdf/2011.09999.pdf) and finding a solution to such non-Markovian constraints would require using a trajectory based formulation that takes into account history (for example, if I’m allowed to only violate one constraint, then I have to remember whether or not I’ve already used up my constraint violation budget in the past when planning future actions and when reasoning about possible constraints when learning). While using epsilon = 0 may seem overly restrictive in stochastic settings, we note that there are multiple prior works which demonstrate that it is possible to learn policies with zero constraint violations even in low-noise stochastic dynamics. Intuitively, this seems possible because agents can act conservatively and create a “buffer” zone so that even in case of noisy transitions, it only ever slips into the buffer zone and never into the constrained states. In highly stochastic settings, however, even this strategy may not work. We think formulating a trajectory-based version of BICRL is an interesting avenue for future work.
>
> **In section 4.1, ...**
>
> This behavior is simply due to random sampling and the stochastic nature of the demonstrations. For this reason, when we present classification results in the paper we average the results over a number of independent runs.
>
>
> **While numerical extensive, ...**
>
> Indeed, in our method our simulations were implemented in discrete state-space environments. Similar to https://arxiv.org/pdf/2109.04874.pdf, we have now added simulations in a continuous state space environment in which we discretize the space in order to implement our Bayesian approach. More specifically, we have added a navigation task in a continuous environment using trajectories from a human demonstrator in section 8. We plan on extending our results in high-dimensional state spaces using preference demonstrations in future work.

---

> > ### Comment · Reviewer_8ZEH · 2022-08-29
> > **Revised manuscript**
> >
> > Thanks for addressing my previous comments. The revised version of the manuscript has much improved the quality of the paper.

---

### Review · Reviewer_Cajr · 2022-07-27

**Summary Of Contributions:**

This paper presents a Bayesian method for inferring state-space constraints from expert state-action trajectories in an MDP whose (nominal) reward and transition function is known. Specifically, the authors assumes that expert act according to a Boltzmann stochastic policy following an optimal Q-function given an augmented reward function consisting of the nominal reward plus a constant penalty term if the current state is in the true constraint set. Given this model, the authors propose a Markov-chain Monte-Carlo approach to perform Bayesian inference on the constraint set as well as the value of the penalty. Furthermore, the authors propose a hierarchical approach to scale the algorithm to larger state spaces, if the larger problem can be broken down into distinct subspaces with independent subtasks which share the same constraints. Finally, the authors also extend the approach to learning constraints which are defined in terms of known features, rather than on the state-space itself. The authors demonstrate the approach through a set of numerical experiments on a set of grid navigation tasks, including grid-world problems, as well as home navigation and a lane change scenario, comparing their approach to alternative Bayesian inverse reinforcement learning algorithms which make fewer assumptions of the reward structure, as well as a non-Bayesian constraint inference algorithm in GICI.

**Broader Impact Concerns:**

I do not think the lack of a Broader Impact statement here is an egregious limitation, but the authors could consider adding a discussion of the implications of both false-positive and false-negative constraint identification.

**Requested Changes:**

Major changes:
- More experiments on demonstrations that don’t exactly follow the modeling assumptions made by BICRL
    - Synthetic data generated with different parameters (e.g. different value of $\beta$, or noisy expert model which respects constraints) *(critical)*
    - Real world demonstrations, e.g. human controlled navigation in the apartment navigation task. *(strengthen)*
- Amended active learning discussion to include what sorts of demonstrations would be most informative in reducing ambiguity about constraints. *(strengthen)*
- Added discussion highlighting the limitations of Hierarchical BICRL, namely that it requires different demonstrates, and discussion of how diversity in data may contribute to the better performance of BICRL. *(critical)*
- Potentially add experiments comparing against Global BICRL with demonstrations from multiple tasks to distinguish the benefits of the small subproblems from the benefits from more diverse data. *(strengthen)*
- Details on how demonstration data in the home navigation task was collected, and what the subtasks were. *(critical)*

Minor changes:

- Section 3.1: “Prior work by Paternain et al. (2019) shows that the problem has zero duality gap and hence the solutions of the aforementioned two problems are equivalent.” What are the two problems? As written, equation (2) is just an objective function, not an optimization problem. *(critical)*
- Section 3.3: “We further allow for randomly accepting proposals even if they are not associated with a higher likelihood to enhance exploration in the Markov Chain.” As far as I can tell, this is standard practice in the Metropolis-Hastings algorithm for MCMC, while the language in the paper suggests it is a unique choice in BICRL. If it is novel, the authors should provide more motivation for this choice, while if it is not, then the authors should make clear that the implementation is the standard Metropolis-Hastings algorithm for MCMC. *(critical)*
- Algorithm 1: “if sample constraints” is unclear, amend to make more explicit that this if clause is a form of scheduling *(strengthen)*
- Figure 2: Subfigure (a) is unrelated to subfigures (b) and (c), perhaps (a) should move to be part of Figure 1. *(strengthen)*
- Section 5: “Stochastic dynamics, except for providing a more general framework to the deterministic ones, can be seen as…” Using “except” in this case is confusing,  reword, perhaps to “beyond providing…” *(strengthen)*
- Figure 8 and Figure 14 are inconsistent - 14 shows false constraints identified near the doorways. These should be made consistent and the potential cause of the false positives should be discussed. *(critical)*
- Add a discussion of future work *(strengthen)*

**Strengths And Weaknesses:**

Strengths:
- The paper is well written and generally well organized, and situated the work well in the literature.
- The proposed algorithm is clean and well justified, and performs well on the proposed experiments.
- The proposed modeling framework yields a simple and easy to implement algorithm.
- The authors do well to highlight the benefits of the uncertainty quantification provided by a Bayesian approach by including an active learning example.

Weaknesses:
- The paper is posed as a method for learning MDPs with hard constraints (having an expected constraint violation equal to 0, as stated in equation (1), but the algorithm is designed only for soft constraints which add a fixed penalty for constraint violation. The authors argue that this is justified, as the penalty formulation arises in the Lagrangian of the original problem. However, while it is true that there exists a value of the penalty $r_p$ such that the solutions to the penalized formulation and the original problem are the same, the specific value of $r_p$ will depend on the nominal reward function, and not necessarily be the same across different tasks in the same environment. For all of the experiments, expert trajectories were drawn by using a Botlzmann exploration policy with a reward function with soft constraints encoded by some true value of $r_p$ (except perhaps the home navigation example, for which details were not given). However, realistically, we cannot know that the experts are acting with this type of constraint model. While assuming this soft constraint model may be a useful modeling assumption, the authors should demonstrate how this approach generalizes to expert trajectories which follow a different model of constraints (such as hard constraints which are never violated in the demonstrations?)

- More generally, I found the experimental comparisons to be limited. As mentioned, the experiments all used expert trajectories that were drawn from the exact model assumed by the algorithm: soft constraints which share an equal penalty, and actions executed via a Boltzmann policy with $\beta = 1$. Within this setup, it is not surprising to me that BICRL outperforms the BIRL, BCPR, and BDPR baselines, as the latter algorithms make fewer assumptions on the reward structure. The paper would be greatly strengthened by experimental evaluation under settings which do not exactly meet the modeling assumptions chosen by BICRL, including real-world data collected from human expert behavior, or data where the “optimality” of the expert data isn’t known to the algorithm, i.e. the $\beta$ that is used to collect data is different from that used in the inference step.

- The active learning approach feels suboptimal: As after each iteration, the expert is only queried on a single state, rather than a collection of the most uncertain states. Furthermore, as the experts aim to avoid constraint states, their behavior when in a constraint state would not necessarily be informative. It seems that sampling states that are in the backward reachable set to states with high uncertainty would provide the most information. The authors should better motivate their approach to the active learning problem, and at the very least highlight this potential limitations of their approach.

- Finally, the paper lacks some important details and discussion in the experimental results:

    - The section on Hierarchical BICRL compares the original BICRL algorithm argues that the hierarchical version yields better constraint estimation and also improved computational efficiency. However, this is not necessarily a fair comparison (even with more demonstrations given to the Global version) — the hierarchical version sees a more diverse set of task demonstrations which give better coverage of the state space. Thus, the two algorithms cannot work on the same dataset — one needs to collect separate subtask data to use the hierarchical version. This distinction should be emphasized, as in real world settings, expert demonstrations may be a scarce resource.

    - The section on the home navigation task is quite short and does not provide enough details about how the experiment was set up. The discussion in the appendix still leaves important details missing: How are the trajectories generated? How were the subtasks defined?

---

> ### Author Response · Authors · 2022-08-26
> **Response to Reviewer Cajr**
>
> We would first like to thank the reviewer for all the comments and suggestions. Below, we address each requested change individually.
>
> **Comment: More experiments on demonstrations that  ...**
>
> We thank the reviewer for this comment. We have added further experiments to showcase the performance of our method when the rationality levels between the demonstrating and the inferring agents are different. In subsection 5.2 we study the impact of having different betas between the demonstrator and the inferring agent. Furthermore, in section 8 we added a simulation in which we provide the inferring agent human generated demonstrations in a continuous state space.
>
> **Comment: Amended active learning discussion to include  ...**
>
> It is true that our active learning approach is an intuitive, but simple one. The active learning approach however is not the primary focus of this work, hence, we have only included a simple approach to elucidate the usefulness of our Bayesian method towards active learning. We have included a note in this regard in the corresponding section of the main paper as well and leave exploration and development of more sophisticated active learning approaches to future work.
>
> **Comment: Added discussion highlighting the limitations of  ...**
>
> We agree with the reviewer that Hierarchical BICRL has limitations. We have added a discussion regarding that in the Hierarchical BICRL section in which we emphasize the fact that indeed Hierarchical BICRL requires different demonstrations for subtasks in comparison to the global case. Furthermore, it is true that having more diverse demonstrations will most likely improve classification as a greater area of the state space is explored.  We have also added some discussion on limitations of Hierarchical BICRL in the future work and  limitations section (Section 10).
>
> **Comment: Potentially add experiments comparing against  ...**
>
> We agree with the reviewer that most likely the performance of Hierarchical BICRL with subtask demonstrations and that of Global BICRL with a variety of demonstrations in the entire domain will have a comparable performance. Our aim in introducing Hierarchical BICRL was to mainly argue that constraints are indeed modular and that they can be inferred locally as well. In case local demonstrations are available, then constraints in sub-domains can still be inferred with a significant computational gain in comparison to the global case.
>
> **Comment: Details on how demonstration data in the  ...**
>
> We have added the details in the Appendix B1. We also changed Figure 10 so that the trajectories shown there, which originally were meant to be just conceptual, are now a more accurate depiction of the tasks taking place. Please note that the numbers on the figures have changed as we have added more results in the sections above.
>
>
> **Minor Changes**
>
> **Section 3.1**
>
> We have changed equation 2 so that it is now an optimization problem.
>
>
> **Section 3.3**
>
> Indeed, this is a standard approach in sampling methods. The sentence now reads “As in the Metropolis-Hastings algorithm for Markov Chain Monte Carlo methods, we also allow for randomly accepting proposals …”.
>
> **Algorithm 1**
>
> We have amended this by including a sampling frequency variable which now makes clear that this is a form of scheduling.
>
> **Figure 2**
>
> We agree and we moved the figure to Figure 1.
>
> **Section 5**
>
> We have changed the wording to “in addition to …”
>
> **Figure 8 and Figure 14**
>
> Indeed Figure 18 (in the first submission this was Figure 14) was not updated to match the results of Figure 10 (formerly Figure 8). Please note that the numbers of the figures have changed as we added more results in the sections above. We also added a discussion regarding the occasional false positives.
>
>
> **Add a discussion of future work**
>
> We have added a section in which we discuss the limitations of our work and possible future directions for improvement.
>
> Finally, please note that we have colored in blue the additional results/changes in the write up for your convenience.

---

> > ### Comment · Reviewer_Cajr · 2022-09-08
> > **Revised Manuscript**
> >
> > Thanks for addressing my comments. The changes in the manuscript address my concerns -- the results on the discrepancy in rationality really improve the work.
> >
> > I just have a few very small issues that I noticed in the updated manuscript:
> >
> > - In Equation (2), the inequality is in the wrong direction under the min operator (it should be $r_p \le 0$). The text should make clear that conventional RL solvers can only be used for fixed values of $r_p$, and not on the min max problem directly.
> >
> > - The ending quotation mark on "risk tolerance" is poorly formatted and does not match the opening quotation mark.

---

> > > ### Author Response · Authors · 2022-09-09
> > > **Response to Reviewer Cajr**
> > >
> > > We would like to thank once more the reviewer for carefully reading our paper and for providing valuable feedback. We addressed the issues in a new revised version of the paper. The updated text is now colored in orange.

---

### Review · Reviewer_a2ZC · 2022-08-13

**Summary Of Contributions:**

This paper considers the problem of inferring unknown constraint states in a constrained MDP with known transition model, nominal reward function, and finite state and action spaces. The dual formulation of the constrained RL objective leads to an MCMC algorithm where both the constraint states and the Lagrange multiplier (reward for violating the constraint states) are sampled. The resulting posterior can also be used for active learning. The paper also proposes feature-based and hierarchical extensions of the main approach. Experiments are in a variety of grid-like domains, including robotic home and highway environments, with comparisons against several baselines.

**Requested Changes:**

* I am curious about the time complexity of the proposed BICRL vs the GICI baseline. My understanding is that GICI does not have to run value iteration in an inner loop, so I imagine it could be significantly faster than BICRL. The comparisons in Section 4.2 are against GICI using constant threshold on KL divergence as a stopping criterion. Could we instead see a graph with wall-clock time on the x axis and performance (either FNR, FPR, or Precision) on the y axis, and one line for each of BICRL and GICI? Or instead, could we see additional results for smaller thresholds for the stopping criterion of GICI, to rule out the possibility that the constant 0.1 was just a poor choice in this domain?
* Can you clarify whether all of the baselines assume a known transition model?
* The second paragraph of Section 3.2 “Problem Statement” contains information about the approach, rather than the problem statement. I would suggest moving this information elsewhere, to let the problem statement stand alone.
* What is the motivation for including results with Expected a Posteriori (EAP)? In a practical application, would EAP ever be a better choice than MAP?
* A consistent grammatical issue is missing commas, especially following introductory clauses. For example:
   * For these simulations we utilized 100 expert trajectories. (There should be a comma after “simulations”)
   * In this section we motivate BICRL… (There should be a comma after “section”)
   * For a large number of environments and tasks safety constraints are compositional. (There should be a comma after “tasks”)
   * There are many others -- please check the first sentence of every paragraph and apply this change everywhere.

## Minor

* “in a Hierarchical manner” → “in a hierarchical manner”
* “an Hierarchical version” → “a hierarchical version”
* My understanding is that “constraint allocation” and “constraint” are used interchangeably. If there is a distinction, please clarify, otherwise, it would be clearer to stick with one (probably just “constraint”).
* The Scobee & Sastry reference should be updated from arXiv 2019 to ICLR 2020.


**Strengths And Weaknesses:**

### Strengths

* I like that the paper considers not only constraint inference, but also active learning and feature-based and hierarchical versions of the approach. Altogether, they comprise a thorough contribution.
* The engagement with previous work is very good throughout the paper, both in terms of the writing and in terms of well selected baselines in experiments.
* Using the Paternain et al. (2019) zero duality gap result as motivation makes sense and provides a good theoretical foundation for the work.
* The writing is overall very clear and well organized.
* The experimental environments are relatively simple and gridworld-y overall, but seems as-good-as or better than previous work in this line (IRL is hard!).


### Weaknesses

* In contrast to previous work, this paper allows demonstrators to sometimes violate constraints. On one hand, this is perhaps more realistic, since demonstrations are often imperfect. On the other hand, if constraints can be violated, it calls into question whether “constraints” are really the appropriate framing. It may be more accurate to describe the setting as one where there is a strong prior over the reward function (each transition either elicits the known nominal reward or an unknown other constant reward). In fact, I think that nothing would change in the approach if rather than the unknown “constraint” states being uniformly bad, they were actually uniformly good, or just uniformly somewhere in between.
* Relatedly, my understanding is that the only difference between BICRL and the ​​baselines BCPR and BCDR is that the former assumes a state-independent reward prior and the latter have state-dependent reward priors.
* As far as I can tell, the demonstrations used throughout this work are collected synthetically by running a policy that conforms to the Boltzmann-type model. It would be much more compelling to show results with human demonstrations, or at least demonstrations that do not directly conform to the model assumed in this work.

---

> ### Author Response · Authors · 2022-08-26
> **Response to Reviewer a2ZC**
>
> First of all, we would like to thank the reviewer for the insightful feedback. Below, we address each of the requested changes individually.
>
> **Comment: I am curious about the time complexity ...**
>
> Both GICI and BICRL run value iteration (feature accrual) in the inner loop. In terms of wall clock time, GICI is more likely to be faster as it is a greedy approach and it does not have to run a MCMC chain. It is also important to note that GICI only recovers a point estimate, however, BICRL provides a complete posterior over different constraint sets. We have changed the simulations in this section in the following way. We now study three different scenaria, one with no noise in the transition dynamics and two with different levels of noise. We further tune the KL divergence criterion of GICI in order to obtain the best possible results each time. We have also added in the Appendix a table with the classification results from GICI for varying levels of the KL divergence stopping criterion.
>
>
> **Comment: Can you clarify whether all of the baselines ...**
>
> Yes, they all assume a known transition model.  We now have also made this clear in the preliminaries section of the updated paper.
>
>
> **Comment: The second paragraph of Section 3.2 ...**
>
> Thank you for this remark. We have moved this paragraph to the next section that introduces our Bayesian method, since it contains information regarding our Bayesian approach.
>
> **Comment: What is the motivation for including results ...**
>
> In most experiments we use the MAP results to quantify the classification performance. However, the EAP values contain information about the uncertainty in the estimation which we believe is  also important. The EAP results also reveal information about the MAP estimates for the states with little to no uncertainty.
>
>
> **Comment: A consistent grammatical issue is ...**
>
> Thank you for pointing this out. We hope we have now addressed most of these inconsistencies in our latest submission.
>
>
> **Minor:** Furthermore, we have addressed the minor changes in the updated version of the paper. We have colored in blue the additional results/changes in the write up for your convenience.

---

> > ### Comment · Reviewer_a2ZC · 2022-08-28
> > **Thanks!**
> >
> > Thanks for addressing my feedback and answering my questions. The paper improvements look good to me, especially the additional GICI results and the experiment with human demonstrations. (The paper is now well over 12 pages, but I'm not sure if this matters for TMLR.)

---

### Author Response · Authors · 2022-08-26
**To All Reviewers**

We would like to thank all the reviewers for their valuable comments and suggestions which we believe helped us improve our paper. Below, we list the main updates in our paper as requested by the reviewers:

*  We have carried out further simulations regarding GICI by appropriately tuning the KL divergence stopping criterion in both deterministic and stochastic environments. These updates can be seen in Section 4.2 and Appendix A.1.

*  We have now included simulations when demonstrations do not follow the modeling assumptions of BICRL. More specifically, in Section 5.2 we study the effects when the temperature parameter $\beta$ used to obtain the demonstrations is different from the value of $\beta$ assumed by BICRL. Furthermore, in Section 8 we have included experiments in which BICRL infers constraints by observing human generated trajectories in a continuous state space environment.

* We have clarified in our write up the limitations of our active learning approach and hierarchical BICRL.

*  Regarding the theoretical properties of BICRL, we have now added a proof of rapid MCMC mixing to Appendix D and show that BICRL inherits the same rapid mixing properties as Bayesian IRL.


Below, we address the comments of each reviewer individually. For convenience, we have colored all the major changes in our updated submission in blue .

---

### Decision · Action_Editors · 2022-09-17

**Recommendation:** Accept as is

**Comment:**

The paper considers the problem of inferring state-space constraints in a known MDP from demonstrations. Contemporary methods primarily focus on maximum likelihood inference of constraints, and thus do not model or utilize the uncertainty in the resulting constraint estimates. In contrast, the paper proposes a Bayesian approach to inferring state-space constraints for an MDP with known transition and nominal reward functions based upon expert state-action trajectories. These trajectories are assumed to correspond to a Boltzmann policy, where the nominal reward is augmented with a fixed reward penalty when a state lies in the constraint set. The proposed algorithm uses MCMC to perform Bayesian inference over the constraint set and the reward penalty. The paper then describe feature-based and hierarchical variations of their algorithm that extend this approach to larger state spaces. As a demonstration of the usefulness of uncertainty estimates for the inferred constraints, the authors present a simple approach to active learning that utilizes the resulting posterior. The paper compares the performance of the proposed constraint inference algorithm to recent Bayesian and non-Bayesian baselines in a series of discrete or grid-like domains, as well as a continuous-state environment.

The paper was thoroughly reviewed by three knowledgeable reviewers, who read the authors' responses and updated their reviews accordingly. The reviewers agree that the final manuscript provides a valuable contribution to RL constraint inference---the proposed Bayesian inference algorithm is novel and theoretically sound, and its effectiveness is experimentally demonstrated on a variety of domains. The reviewers also agree that the paper is well written and clearly organized, and that it provides a solid discussion of and comparison to existing work on constraint inference. The reviewers raised several key concerns about the initial submission, notably: the fact that the experiments involved synthetically generated demonstrations that conform to the modeling assumptions by construction; questions about the comparisons to the GICI baseline; and limitations of the hierarchical variation of the approach and its use for active learning. The authors clearly put considerable effort into addressing the reviewers' questions and concerns, which includes carrying out additional experiments with non-conforming demonstrations and updated implementations of GICI, and revisions the text to speak to limitations. All three reviewers appreciate these changes and agree that the final manuscript warrants acceptance.